# LBA-YOLO: A novel lightweight approach for detecting micro-cracks in building structures

**Wenhao Ren, Zuowei Zhong**[ID]¤*

School of Civil Engineering, Inner Mongolia University of Technology, Hohhot City, China

¤ Current address: Hohhot, Inner Mongolia, China
* nmggydxzzw@126.com

## Abstract

Developing an efficient and accurate algorithm for detecting building cracks, especially micro-cracks, is essential for ensuring structural integrity and safety. The identification and precise localization of cracks remain challenging due to varying crack sizes and the inconsistency in available datasets. To address these issues, this study introduces an innovative crack detection model based on YOLOv8n. The proposed method incorporates two novel components: AC-LayeringNetV2, a hierarchical backbone network that optimizes feature extraction by integrating local, peripheral, and global contextual information, and RAK-Conv, a convolutional module that combines an attention mechanism with irregular convolution operations to enhance the model's ability to handle complex backgrounds. These innovations significantly improve semantic segmentation accuracy while reducing computational overhead. Experimental results on a benchmark dataset demonstrate a 2.20% improvement in precision, a 3.50% increase in recall, and a 1.90% rise in mAP@50 compared to the baseline model. Additionally, the model achieves a 6.55% reduction in size and a 0.03% decrease in computational complexity. These results highlight the practical applicability and efficiency of the proposed approach for automatic crack detection in building structures, emphasizing the novel integration of feature fusion and attention mechanisms to address challenges in real-time and high-accuracy detection of micro-cracks in complex environments.

## 1. Introduction

A comprehensive detection system is essential for the safety evaluation of civil infrastructures. Timely identification as well as subsequent maintenance of civil infrastructures are essential to mitigate damages as well as significant losses in economic terms and human life [1]. Advanced technologies are utilized to facilitate effective damage assessment, ensuring the integrity of civic infrastructures while minimizing, if not eradicating, losses in human lives and financial resources [2]. This constitutes a significant domain for contemporary study, as conventional manual visual inspection

**Data availability statement:** The dataset is publicly accessible under the following DOI: https://doi.org/10.6084/m9.figshare.28540514. We sincerely appreciate any interest in our work, and we invite colleagues to access and utilize these data for further research. Should there be any questions regarding the dataset, we kindly encourage interested parties to contact us for additional information.

**Funding:** Zuowei Zhong Research Program for Science and Technology of Inner Mongolia Autonomous Region in University (NJZY22388) Zuowei Zhong Basic Research Program for Directly Affiliated Universities in Inner Mongolia Autono-mous Region (JY20230007). The funders had no role in study design, data collection and analysis, decision to publish, or preparation of the manuscript.

**Competing interests:** The authors have declared that no competing interests exist.

techniques employed to evaluate the structure integrity of primary infrastructures exhibit several constrains [3].

Dependable, rapid, and effective crack detection techniques are essential for assessing the integrity of structures, as they influence their safety as well as longevity [4]. The outcomes of conventional (manual) crack detection techniques are significantly influenced by the proficiency as well as methodologies employed by the investigators. Manual inspection involves analyzing cracks, namely their location and widths, resulting in subjective outcomes that depend on the inspector's expertise [5]. These constraints result in inadequate damage evaluation of vital infrastructure [6]. Hence, there is a pressing necessity for automated ways to effectively identify cracks in civil infrastructure, thereby surmounting the constraints of manual approaches.

To solve the difficulties of picture segmentation as well as crack identification, many computer vision techniques have been used [6]. Numerous crack detection algorithms make use of various edge detection approaches, like the Sobel and Canny operators, rapid Fourier transform, as well as fast Haar transform [7]. Image contrast as well as background uniformity exhibit a major influence on the effectiveness of basic edge detection methods [8]. To elevate the automated identification of cracks in concrete images, hybrid algorithms are employed [9].

An effective method for automated damage identification is image processing. Image processing approaches facilitate the identification of cracks from the crack picture dataset and allow for essential metrics, including direction and breadth [10,11]. Numerous investigations utilizing image processing methodologies are documented in the literature for the damage identification in concrete structures, encompassing fractures [12,13], as well as damage to asphalt pavements [14]. The image processing algorithms are often adequate and efficient for identifying fractures in certain photos. Nonetheless, its resilience is compromised via the presence of many objects, including light, shadows, rough surfaces, as well as other disturbances encountered in real-world scenarios [15].

To improve the accuracy and efficiency of crack detection, many studies have combined machine learning (ML) with image processing techniques [16–19]. In this process, ML identifies cracks and other structural damages from images through feature extraction techniques [20]. Compared to traditional methods, Deep Learning (DL) is able to automatically extract features from raw data, reducing the need for manual feature engineering, and therefore providing greater performance in crack detection in complex environments.

In deep learning frameworks, target detection algorithms can be broadly classified into two categories: two-stage detection algorithms and one-stage detection algorithms. Two-stage detection methods are able to finely localise targets by generating candidate frames, followed by classification and regression, but they have high computational complexity and a large number of parameters due to the inclusion of two models [3,20–23]. For example, Swarna et al [24] improved the accuracy of crack recognition using ResNet-50 CNN and curvilinear waveform transform.Deng et al [25] proposed a variable module based R-CNN model, while Li et al [26] introduced an attention mechanism to enhance the effect of Faster R-CNN. These methods have

improved in accuracy, but they also face the challenges of parameter storage and transmission, especially in bandwidth-constrained environments.

In contrast, one-stage detection models (e.g., SSD and YOLO) merge classification and localisation into a single step with higher real-time performance and lower computational complexity [27,28]. YOLO is particularly suitable for scenarios that require a fast response, such as bridge crack identification [29–31]. Liao et al. [32] improved YOLOv3 and combined it with K-Means clustering to optimise the anchor frame size, and Cai et al [33] implemented a lightweight detection network by integrating deeply differentiable convolution and attention mechanisms. These innovations improve the detection speed and accuracy, especially for real-time applications.

Further studies such as Tan et al [34] achieved accuracy improvements in the combination of YOLO and UNet3+, or by integrating the ResNet module in YOLOv5. Yu et al. [35] also explored innovative approaches for scalable optimization frameworks in other domains, such as green concrete mix design, emphasizing the importance of integrating advanced technologies into sustainable building practices. Similarly, Liu et al. [36] improved YOLOv5 by reducing memory usage, making it more suitable for real-time detection of small devices. Although YOLOv8 improves on several aspects, it still needs to be optimized for multi-scale target detection in complex contexts [37].

In recent years, deep learning has demonstrated excellent performance in crack detection tasks and overcame the problem of low detection accuracy of traditional methods in complex environments. However, in the field of civil engineering, in addition to deep learning, intelligent computational methods such as Artificial Neural Networks (ANN), Fuzzy Inference Systems (ANFIS) and meta-heuristic optimization algorithms have been widely used to predict the mechanical properties of concrete structures. For example, Ahmadi and Kioumarsi (2023) [38] predicted the modulus of elasticity of high-strength concrete using a combination of ANN and PSO, which efficiently models and optimizes the parameters to improve the prediction accuracy. Meanwhile, Kontoni and Ahmadi (2024) [39] investigated the application of ANFIS-PSO model for axial strain and peak stress prediction of FRP-reinforced concrete, and the results showed that the computational method incorporating intelligent optimization was able to achieve good accuracy and generalization ability in construction engineering applications. In addition, Ahmadi et al. (2024) [40] further used a bio-inspired meta-heuristic algorithm to optimize the compressive strength prediction of high-strength concrete structures, and these studies collectively show that intelligent computational methods have a wide range of potential applications in civil engineering. Although these methods are mainly used for structural parameter prediction rather than crack detection, they validate that intelligent computational methods can effectively improve the automation and accuracy of civil engineering tasks, which is consistent with the goal of this study to optimize crack detection using deep learning. Therefore, this study adopts the YOLO architecture and combines multi-scale feature extraction and optimized IoU computation to further enhance the accuracy and computational efficiency of crack detection.

This study proposes an optimisation model aimed at improving the performance of deep learning-based object detection for non-destructive crack assessment in building structures. The YOLOv8n model is selected due to its fast processing and real-time functionality, offering significant improvements over CNN-based two-stage detectors in speed and accuracy. We introduce a feature optimization module that includes GSConv and GS bottleneck components to enhance feature fusion and processing efficiency. These modules reduce the model's complexity while maintaining its effectiveness. Modifications to the YOLO network's neck structure incorporate the gather-distribute (GD) mechanism to improve feature fusion. Additionally, connections between layers with higher sampling rates are created to preserve smaller target features, thus improving the detection of fine cracks. To further enhance model performance, we integrate the Wise-IoU loss function, accelerating convergence and improving the detection efficiency for complex and fine cracks in construction materials.

## 2. Preface to the work

### 2.1. Yolov8 introduction

YOLOv8 [41] represents the newest addition to the YOLO series of detection algorithms, which covers five architectures: YOLOv8n, YOLOv8s, YOLOv8m, YOLOv8l, as well as YOLOv8x, each optimized for datasets of varying scales. To satisfy

the requirement for instantaneous performance, YOLOv8n is selected as the baseline model. This model achieves better performance while maintaining fast speed and optimizing speed to the extreme. It mainly contains four parts: input, backbone, neck, as well as head, as displayed in Fig 1.

Input: This component handles tasks such as changing the inputs to the required size and performing data preprocessing and augmentation operations. The former one involves normalizing and resizing images to ensure uniform input dimensions and pixel ranges. For data augmentation, methods like resizing, hue modification, mosaic enhancement, as well as casual alterations such as trimming, spinning, and mirroring are used. Additionally, the model utilizes an anchor-free approach to directly estimate object centers, thereby simplifying complication and minimizing reliance on predetermined anchor point dimensions and configurations.

Backbone: The backbone comprises Conv, C2f, and SPPF modules to enhance its feature extraction capabilities. The innovative C2f design leverages residual information for learning, enhancing gradient flow data. Meanwhile, SPPF (spatial pyramid pooling fusion) converts characteristic representation of varying dimensions into consistent-size feature vectors.

Neck: This part follows feature pyramid network (FPN) [42] as well as path aggregation network (PAN) [43], remarkably integrating information flow from hierarchical and foundational pathways within the network, improving effectiveness.

Head: The head part applies feature maps of various sizes to access details on the classification and locations of objects at different scales. By utilizing the distribution focal loss (DFL) [44], the parameter counts and computation complexity is greatly reduced. YOLOv8n achieves big progress in timely detection and shows substantial improvements in precision.

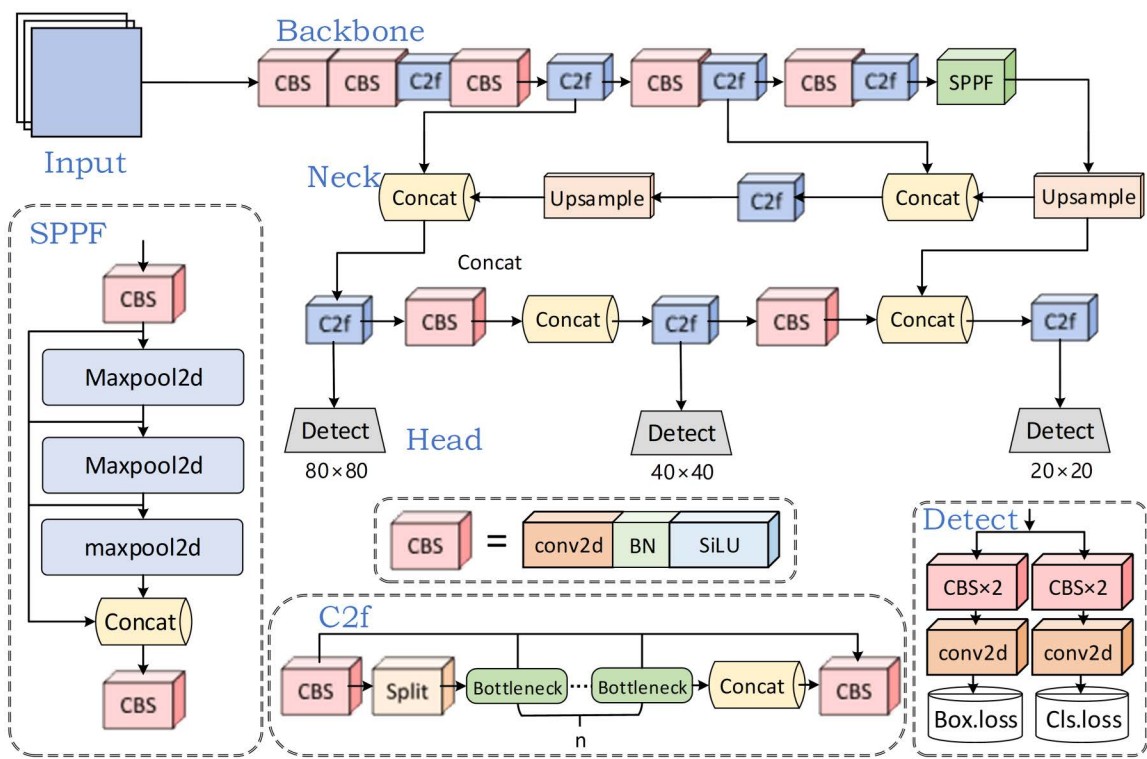

**Fig 1. Structure of YOLOv8n.**

## 2.2. Test introduction

This paper proposes an enhancement to the YOLOv8n target detection algorithm, which is first tested on the 'CityWaste' dataset, and then compared with an accessible open-source image database and an unmanned aerial vehicle (UAV) that collects images depicting mid- and high-rise structural cracks in the city of Hohhot. The results show an improvement in detection efficiency. The proposed enhancement introduces the AC-LayeringNetV2 architecture, which is built on top of the HGNetV2 model. This architecture effectively utilises local features, surrounding environment as well as global context to elevate the precision of semantic segmentation. It significantly lowers the model's computational complexity by employing a hierarchical feature extraction approach.

The following are the study's contributions:

1. A novel RAK-Conv module is introduced, which incorporates the AKConv attention mechanism. This module adapts the convolution kernel to arbitrary parameter counts as well as sampling geometries, focusing on relevant information and minimising attention to irrelevant data. The use of irregular convolutions facilitates efficient feature extraction, elevating the model's capacity to process complicated backgrounds.

2. A new alternative to CIoU is proposed. In addition to calculating the Intersection over Union (IoU), this method considers the outer boundaries of rectangles, computing the minimum distance between them. This provides a more detailed similarity measure for cases in which boundaries are close but not overlapping, providing a finer evaluation than the traditional IoU metric.

3. In addressing the challenge of YOLOv8n' reliance on the quality as well as diversity of training data, this model incorporates several strategies. Firstly, advanced data augmentation techniques, like Mosaic, MixUp, as well as CutMix, are employed to elevate data diversity and improve the model's generalisation. Furthermore, techniques like random erasure, colour jittering, and geometric transformations augment the dataset. Secondly, the employment of synthetic images, generated through the utilisation of computer graphics technology and Generative Adversarial Networks (GANs), enables the model to effectively handle scenarios and targets not present in the original data. Thirdly, a model that has already been trained on large datasets is adjusted for particular purposes, thereby reducing the reliance on substantial amounts of labelled data. Fourthly, a combination of both labelled and unlabelled data is employed during the training process, incorporating pseudo-labeling methodologies to augment the effective training data. Furthermore, the employment of advanced regularization techniques, such as DropBlock and label smoothing, has been shown to enhance model generalization and mitigate the effects of overfitting. Finally, a continuous learning strategy is employed, allowing the model to adapt to evolving data distributions. The integration of these techniques significantly reduces the dependency of YOLOv8n on high-quality, diverse training datasets, improving the model's generalization and robustness.

## 3. Methods

This study did not require a specific license as all data were collected in publicly accessible areas of Hohhot City and did not involve any protected or restricted areas. In addition, the data collection methods used in this study complied with relevant academic and ethical norms and did not fall under any category requiring specific regulatory approval. Therefore, this study complied with applicable research regulations at both the legal and ethical levels.

YOLOv8 has been identified as one of the leading models in object detection. However, as the YOLO family continues to evolve, remarkable advancements have been made in the field, providing a framework for future research. Nevertheless, the primary focus of this research is on YOLOv8. The original YOLOv8 model exhibits certain limitations when confronted with the specific challenges associated with building crack detection tasks, particularly in diverse lighting conditions, complex backgrounds, and the variability in crack morphology. The aforementioned characteristics render the detection of building cracks a more complex and challenging endeavour.

To solve the aforementioned problems, this research proposes targeted enhancements to the architecture of YOLOv8, with the objective of enhancing its suitability for crack detection tasks. Specifically, the enhanced network architecture AC-LayeringNetV2, which is based on HGNetV2 [45], is utilised as the primary network of the model, substituting the standard network structure employed in the original YOLOv8. The efficacy of this enhanced network in extracting local details, surrounding context and global structural features of cracks has been demonstrated, thereby enhancing the accuracy of detection and significantly reducing the computational cost while maintaining performance. Furthermore, this study introduces an attention module, termed RAK-Conv, which facilitates the network's focus on the key feature regions of the crack, especially in cases in which crack edges are blurred or the background is complex. The deployment of this module enables the model to identify the location and shape of the crack with greater accuracy.

This study proposes an enhanced bounding box regression approach to supersede the conventional CIoU [46] method. The novel approach, underpinned by the calculation of IoU, incorporates an additional consideration of the outer boundaries of two bounding boxes, thereby ensuring effective management of scenarios where crack edges are in close proximity yet not fully overlapping. This is achieved by determining the minimum distance between the boundaries. Compared with the conventional IoU method, the proposed approach facilitates a more comprehensive similarity evaluation, consequently enhancing the precision of crack detection.

The enhanced YOLOv8 model, as a result of the aforementioned optimisation measures, demonstrates notable advantages in the domain of building crack detection. These advantages encompass not only an improvement in detection precision but also a substantial enhancement in training efficiency. The model's robustness in complex environments is ensured by these improvements, providing an efficient and reliable solution for the intelligent detection of building cracks.

### 3.1. Lightweight backbone network AC-LayeringNetV2

To address the issues of high computational cost as well as poor real-time performance of traditional methods in the process of building crack detection, AC-LayeringNetV2 network structure was proposed. Compared with YOLOv8 which mainly relies on 3 × 3 convolution operations, AC-LayeringNetV2 significantly reduces the computational cost while maintaining high accuracy, which is suitable for resource-constrained environments. Its architecture mainly includes three sections: initial feature extraction module, multi-stage feature extraction module as well as context enhancement module. This is displayed in Fig 2.

The initial module (StemBlock) completes the preliminary feature extraction through convolution, batch normalization (BN), activation function and pooling operations to reduce the input resolution and computational load, while extracting low-level features like edges as well as textures of building cracks. The multi-stage feature extraction module

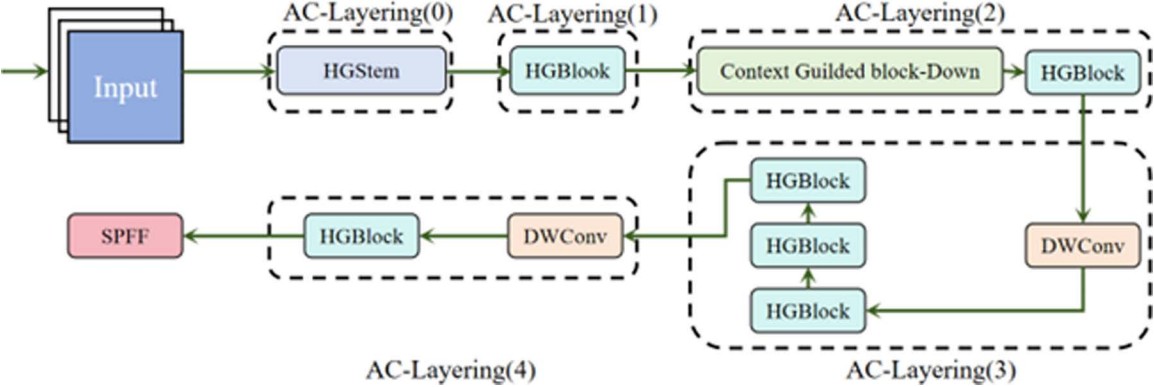

**Fig 2. AC-LayeringNetV2 architecture module.**

(AC-LayeringNetV2 stage) extracts deep features layer by layer. The output of each stage can not only be used as the input of the next stage, but also be used as multi-scale features to detect cracks of various scales. By adding global information, the context improvement module, which is integrated into the first stage, elevates the model's capacity to adapt to changes in the shape and location of cracks.

AC-LayeringNetV2 combines various advanced designs such as Ghost convolution, CSP structure and context guidance module. The input picture is firstly passed through StemBlock for feature extraction and sequentially through 4 HG stages, each of which contains multiple HG blocks. The context enhancement module is introduced after the first stage, which further improves the context awareness of the model by capturing local features, surrounding environment and global background information. Finally, the feature map is passed through the detection head to generate the crack detection result. Its structure is shown in Fig 3.

The architecture design of AC-LayeringNetV2 is inspired by CGNet. By imulating the dependence of human vision on context information, the connection between local and global environment is established in the network, to elevate the accuracy, stability as well as generalization capacity of the model. Its hierarchical feature extraction approach can learn complicated patterns at various scales as well as levels of abstraction, which is especially suitable for tasks that require high precision as well as real-time performance such as building crack detection. Its structure is displayed in Fig 4.

Another improvement proposed in this study is the introduction of a lightweight convolutional architecture, AC-LayeringNetV2, where the core module is LightConvBNAct. This module employs a two-step convolution process to achieve a decrease in the number of parameters and an increase in computational efficiency. The first step adjusts or extends the feature dimensions by 1 × 1 convolution while avoiding the use of activation functions, thus significantly lowering the number of parameters while keeping the spatial dimensionality of the feature map. In the second step, group convolution is used to extract spatial features, and each output channel is convolved via the corresponding input channel to achieve the effect of deep convolution. The use of group convolution effectively decreases the computational overhead, and lowers the number of model parameters.

Assume that the convolution kernel is square and the dimension is $K$, $C_{in}$ denotes the number of the input feature map channels, Cout denotes the number of channels of the output feature map, $H_{out}$ and $W_{out}$ denote the height and width of the output feature map, and $H_{in}$ and $W_{in}$ denote the height as well as width of the input feature map, the computational complexity of the standard convolution can be expressed as follows:

Under the discrete condition, the computational formula of 2D convolution can be further expressed as:

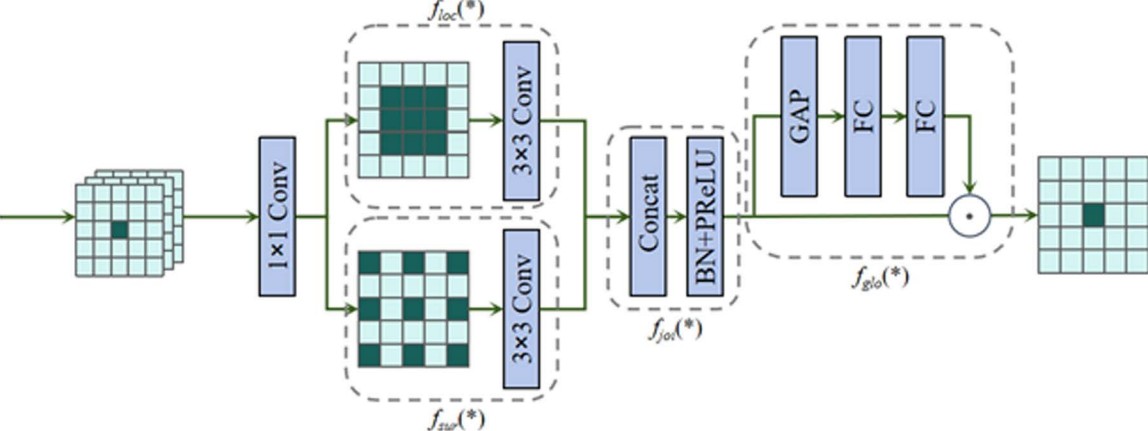

**Fig 3. The structure of the context guided block.**

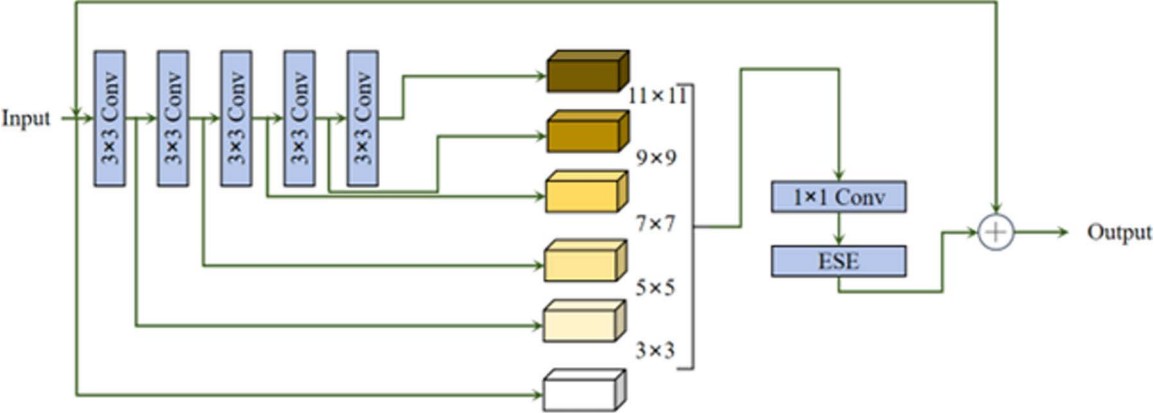

**Fig 4. HG module schematic.**

$$(K * I)(i,j) = \sum_{m=0}^{P-1} \sum_{n=0}^{Q-1} I(i-m, j-n)K(m,n)$$

(1)

Assume that the input matrix and kernel matrix are I as well as K, respectively. The kernel matrix $K$ has shape m × n and its width as well as height are denoted by P and $Q$. To compute the element at position ($i$,) in the output matrix O, the kernel matrix K is slid to the corresponding position of the input matrix $I$. In each step, the corresponding elements of the input matrix $I$ and the kernel matrix $K$ are multiplied point-by-point and all the products are added together to obtain the output value at (i,j).

$$y_l = f(W_l * x_l + b_l)$$

(2)

$$f(x) = \{x \ if \ x > 0; \ \alpha \left(e^{x-1}\right) \ if \ x \leq 0\} \ (\alpha > 0)$$

(3)

Here, $y_l$ denotes the output of layer $l$; $x_l$ denotes the input data; $W_l$ denotes the convolution kernel used in the layer; $b_l$ denotes the bias term; $f$ is the activation function.

$$Calculation \ amount = K \times K \times C_{in} \times H_{out} \times W_{out} \times C_{out}$$

(4)

The computational cost of lightweight convolution can be split into two parts, of which the computational effort of 1 × 1 convolution is one.

$$Calculation \ amount = C_{in} \times H_{in} \times W_{in} \times C_{out}$$

(5)

$$Calculation \ amount = C_{out} \times H_{out} \times W_{out} \times K \times K$$

(6)

By using equations (1)–(6) above it is possible to make the combination of lightweight convolution and context-guided block (CGB) effectively reduces the parameters and optimises the network structure through 1 × 1 convolution and group convolution.1 × 1 convolution merges and decreases the feature channels, thus reducing the computational complexity, while group convolution further lowers the computational cost by processing the feature map groups independently. The

design decreases the computational burden as well as the model parameters, maintaining the expressiveness of the network by fusing local as well as global features through CGB, thus achieving better performance in resource-constrained environments.

## 3.2. Effective attention mechanism

The main difficulty in addressing the challenges in building crack recognition is to overcome the difficulties posed by the diversity and complex distribution of crack sizes. Conventional convolutional operations suffer from two problems. Firstly, these operations are limited to a local receptive field, cannot effectively acquire long-range information, and have a fixed sampling architecture. Second, square convolution kernels and fixed sampling configurations are poorly adapted to dynamically changing targets. To address these issues, our approach integrates the AKConv [47] adaptive convolution kernel mechanism into the model. AKConv dynamically determines the convolution kernel coordinates through an innovative algorithm and adjusts the sampling positions using variable offsets to better adapt to crack changes. This dramatically reduces model computation and storage overheads while improving detection accuracy.

In addition, we use the Mish [48] activation function instead of the SiLU activation function in AKConv to further optimise performance. After the convolution operation, the SEnet [49] module was added to recalibrate the channel importance of the output feature maps of the convolutional layer to ensure that the model can better utilise the feature information. Ultimately, the new module was named RAK-Conv and the module's workflow is displayed in Fig 5.

These improvements remarkably elevate the accuracy as well as efficiency of building crack detection.

The dimensions of the input image are (C, H, W), in which C is the number of channels; H and W are the height as well as width of the picture. AKConv delivers an initial sample shape for the convolution kernel and uses the learnt displacements to adjust the sample shapes after performing the Conv2d operation. The feature maps are resampled, reshaped, convolved and normalised, followed by output via the Mish activation function and finally optimised via the SEnet module.

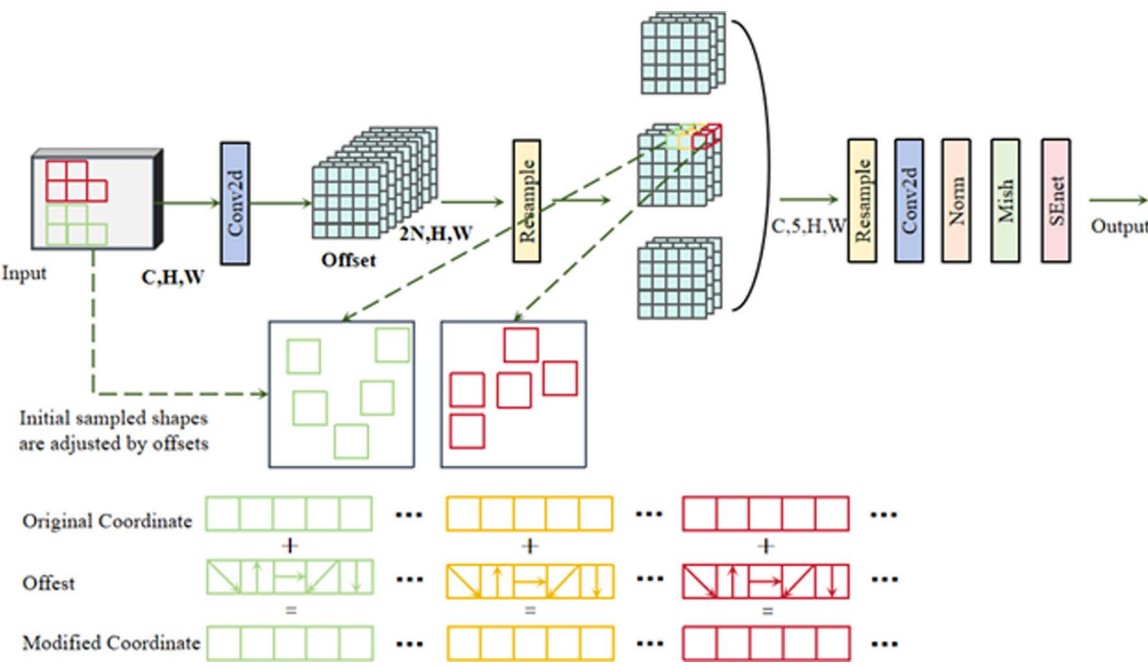

**Fig 5. RAK-Conv.**

Next, the paper details the derivation process of the attention mechanism and defines the associated functions. As shown by the following equations (7)–(13).

$$g(X) = W_{n+1} * \mathrm{Re}\,LU(W_n * X + b_n) + b_{n+1} \tag{7}$$

$$W_1 = W_1 - learning\_rate * \partial/\partial W_1 \tag{8}$$

$$b_1 = b_1 - learning\_rate * \partial/\partial b_1 \tag{9}$$

$$W_x = W_x - learning\_rate * \partial/\partial W_x (x > 1) \tag{10}$$

$$b_x = b_x - learning\_rate * \partial/\partial b_x (x > 1) \tag{11}$$

$$W_n = W_n - learning\_rate * \partial/\partial W_n \tag{12}$$

$$b_n = b_n - learning\_rate * \partial/\partial b_n \tag{13}$$

The loss function is denoted as $L$, where bn denotes the effect of each element change on the loss and $W_n$ is the contribution of the cell change to the total loss. Calculated at the nth level, the partial derivative of the loss function with respect to the weight matrix $W_n$ is $\partial L/\partial W_n$. This process is repeated until the model converges. Through this process, the weights $W_n$, $W_{n+1}$ as well as the biases $b_n$, $b_{n+1}$ are gradually adjusted to their optimal values, allowing the attention mechanism to concentrate on key features. As shown by the following equations (14) and (15).

$$\alpha(X) = soft\max(g(x)) = \frac{\exp(g(x))}{\sum \exp(g(X))} \tag{14}$$

$$X' = \alpha(X) \odot X \tag{15}$$

The input feature $X$ is a multidimensional tensor, the function $g(X)$ is a small neural network to compute the attention score, and $a(X)$ is the computed attention weight. The weighted output features are notated as $X'$, where $\odot$ denotes the element-level multiplication (Hadamard product).

Compared with traditional convolution methods, RAK-Conv provides more choices by linearly increasing the convolution parameters through the kernel size, thus effectively reducing the number of parameters and computational cost of the model. The parameters of traditional convolution grow quadratically with kernel size. As shown by the following equations (16)–(18).

$$P_{traditional} = K^2 \times C_{in} \times C_{out} \tag{16}$$

$$P_{traditional} = (K^2 \times C_{in} + 1) \times C_{out} \tag{17}$$

$$P_{linear} = N \times C_{in} \times C_{out} \tag{18}$$

Following the convolution operation, the SEnet module was added to elevate the model's understanding of the dynamic relationships between channels. The enhancement is based on the high-level features provided by AKConv, optimising the the model's overall performance. The approach dynamically adjusts the convolution kernel position through a spatial attention mechanism, elevating the model's capability in processing channel information. The SEnet module's specific structure is displayed in Fig 6.

The feature map with the RAK-Conv attention mechanism can be represented by equations (19) and (20). $x$ is the initial input feature map, and $X_{offset}$ denotes the feature map adjusted by positional offsets as well as bilinear interpolation. $conv(X_{offset})$ is the rearrangement as well as convolution operations performed on the adjusted feature map. mish is the activation function applied to the convolved feature map, and SE stands for the SEnet module application. The final output feature map is noted as $X'$.

$$Mish(X) = Xtach(\ln(1 + e^x)) \tag{19}$$

$$\widetilde{X} = SE(Mish(Conv(X_{offest}))) \tag{20}$$

## 3.3. The improved loss function MPDIoU

The base YOLOv8 technique uses bounding box regression with the CIoU loss function. Although CIoU is effective to some extent, it has some limitations. Firstly, CIoU does not have a good mechanism for resolving the differences between complex and simple samples, which is the key to elevating the robustness of the model. Second, although CIoU includes aspect ratio as a penalty term in its formula, it cannot accurately reflect the differences between predicted and actual bounding boxes, especially when these boxes have the same aspect ratio but various sizes. In addition, the computational process of CIoU involves inverse trigonometry operations, which increases the computational burden of the model.

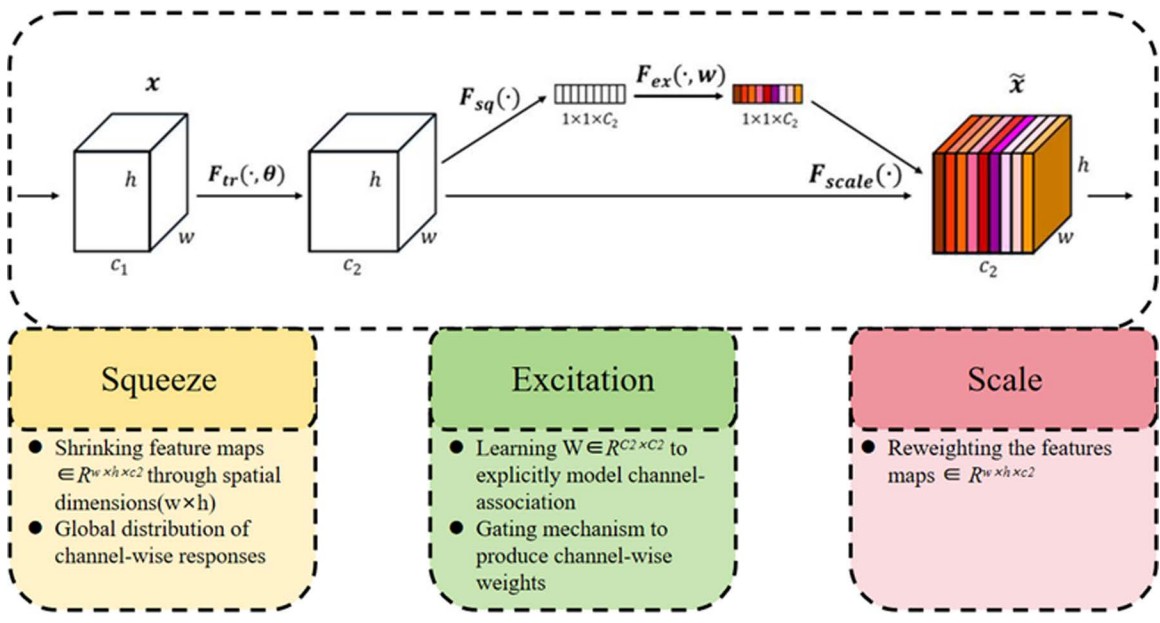

**Fig 6. SEnet module.**

 

$$L_{CIoU} = 1 - IoU + \frac{\rho^2(b, b^{gt})}{(c_w)^2 + (c_h)^2} + \frac{4}{\pi^2}\left(\tan^{-1}\frac{w^{gt}}{h^{gt}} - \tan^{-1}\frac{w}{h}\right) \tag{21}$$

The intersection and concatenation (IoU) measures the proportion of the overlapping area between the prediction frame and the real frame to their joint area. Fig 7a illustrates the relevant parameters in the formulation. $\rho$, $b$, and $b_{gt}$ denote the Euclidean distance between the centre of mass of the prediction frame as well as the real frame, h and w are the height as well as width of the prediction frame. $h_{gt}$ and $w_{gt}$ are the height as well as width of the real frame. $c_h$ and $c_w$ are the dimensions of the smallest rectangular box surrounding the prediction frame as well as the real frame.

In Fig 7b, entities A and B represent the predicted as well as real bounding boxes. The coordinates of the upper-left as well as lower-right corners of bounding box A are $(x_1^A, y_1^A)$ and $(x_2^A, y_2^A)$, while the corresponding coordinates of bounding box B are $(x_1^B, y_1^B)$ and $(x_2^B, y_2^B)$. The spatial separation between the top left and bottom right corners of the real and anticipated frames is represented by the variables $d_1$ and $d_2$.

$$L_{MPDIoU} = 1 - \frac{A \cap B}{A \cup B} - \frac{(x_1^B - x_1^A)^2 + (y_1^B - y_1^A)^2}{w^2 + h^2} - \frac{(x_2^B - x_2^A)^2 + (y_2^B - y_2^A)^2}{w^2 + h^2} \tag{22}$$

$$GIoU = IoU - \frac{|C - (A \cup B)|}{|C|} \tag{23}$$

$$DIoU = IoU - \frac{\rho^2(b, b^{gt})}{(c_w)^2 + (c_h)^2} \tag{24}$$

$$\frac{\partial MPDIoU}{\partial P} = \frac{\partial IoU}{\partial P} - \alpha * \left(\frac{\partial MPD}{\partial P} - MPD * \frac{\partial c}{\partial P * c^2}\right) \tag{25}$$

From equations (21)–(25), it can be seen that it improves the accuracy of bounding box regression in MPDIoU compared to the standard IoU by introducing a penalty in the loss function for differences in box sizes. This improvement is able to better handle the error between boxes of different sizes, overcoming the limitations of traditional IoU in this case. By optimising the computational process, MPDIoU is able to accelerate the convergence of the model while elevating the regression precision.

Compared with GIoU and DIoU, MPDIoU pays more attention to the difference in perimeter of the bounding box rather than just considering the geometry or centre distance of the box. Unlike GIoU, which deals with the difference of bounding boxes through the minimum outer bounding box, MPDIoU uses the minimum perimeter distance to measure the difference between the boxes, which has better adaptability and especially performs more sensitively when dealing with irregularly shaped targets.

In terms of gradient computation, MPDIoU makes the relationship between IoU and perimeter difference adjustable by introducing a balancing parameter, which further optimises the computation process of the loss function and improves the stability as well as performance of the model in various scenarios.

## 4. Experiments

### 4.1. Dataset

The initial step of the present investigation involved the collection of pictures depicting cracks in mid- to high-rise structures, specifically those with 5 or more stories, located in Hohhot, utilizing accessible open-source image databases as well as unmanned aerial vehicles (UAVs). The utilization of UAVs for data collecting minimizes disruption to standard

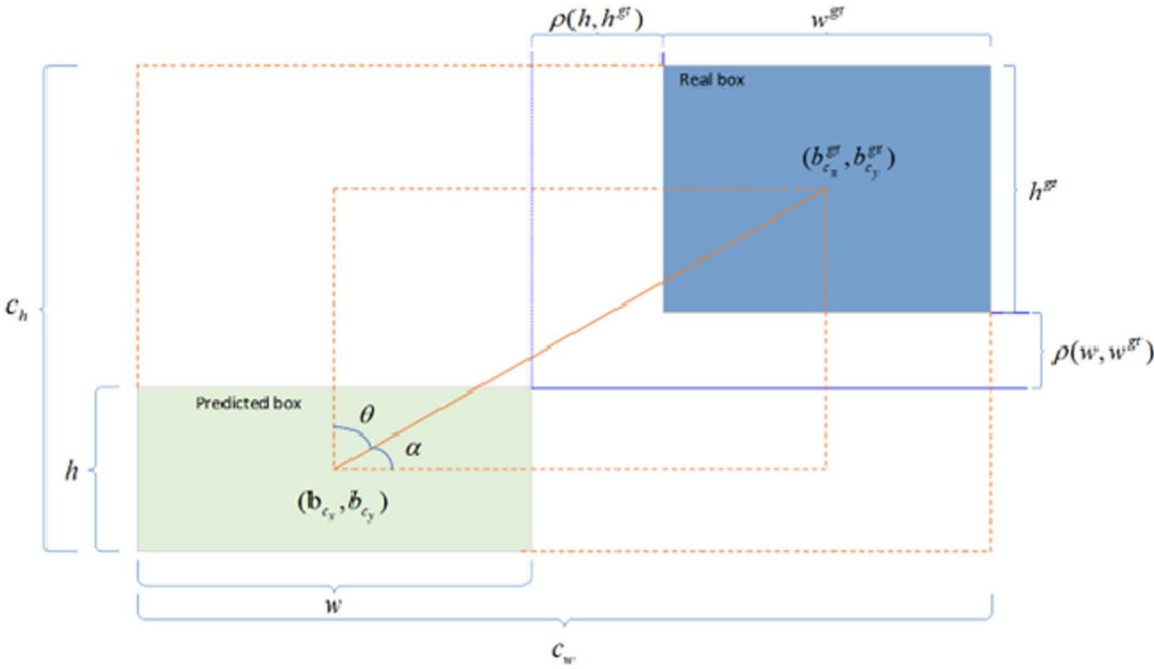

## (a) Loss function-parameters

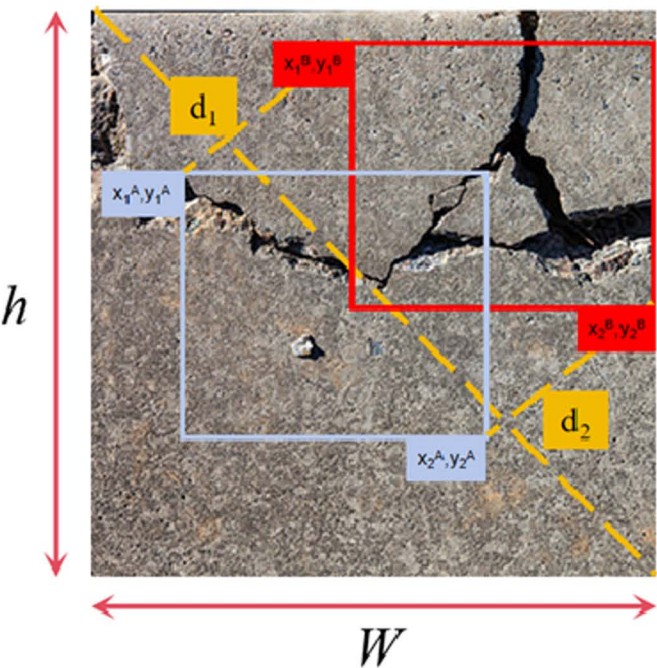

## (b) MPDIoU parameter diagram

**Fig 7. Loss function comparison plot.**

building operations and offers adaptable, economical methods for obtaining photos of cracks. Fig 8 displays photos of the cracks examined in this study at the sites (several buildings in Hohhot).

The crack picture set was obtained via the DJI-M200 quadcopter, which has a GNSS receiver, barometer, inertial measurement unit (IMU), as well as the ability to take off and land vertically (VTOL). 800 crack images in all were acquired to be included in the finished dataset.

To satisfy experimental criteria, this research randomly divided the dataset into training as well as validation sets in an 8:2 ratio. It was annotated with a tagging tool and concentrated on a particular category. As seen in Fig 9, annotations for the training set contained width and height measurements, category names, and the center coordinates (x, y) of the bounding boxes.

## 4.2. Experimental environment and assessment indicators

An Intel Xeon CPU E5-2680 v3 as well as an NVIDIA GeForce RTX 2080 Ti GPU with 11 GB VRAM were used in the Linux environment for the research. PyTorch 1.7.0 as well as Python 3.8 were utilized as the frameworks. The following hyperparameters were set up for the experiment: a batch size of 32, a weight decay coefficient of 0.0005, a momentum of 0.937, 150 training epochs, as well as an initial learning rate of 0.01.

Four main metrics—Precision (P), Recall (R), F1-score (F1), as well as Mean Average Precision (mAP)—were employed to precisely evaluate object detection ability. The following shows the precise formulas:

$$P = \frac{TP}{TP+FP} \tag{26}$$

$$R = \frac{TP}{TP+FN} \tag{27}$$

$$F1 = \frac{2*P*R}{P+R} \tag{28}$$

$$mAP = \frac{\sum_{q=1}^{Q} AP(q)}{Q} \tag{29}$$

"TP" stands for genuine positives, or accurately detected positive samples, in the calculations stated above. "FP" is false positives, which are negative samples mistakenly labeled as positive. "FN" is false negatives, which are positive

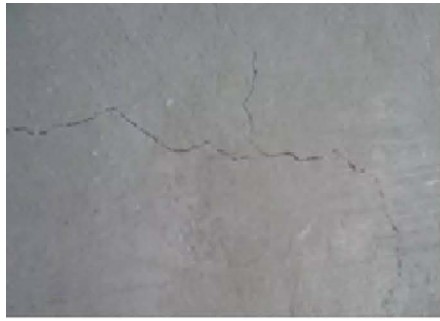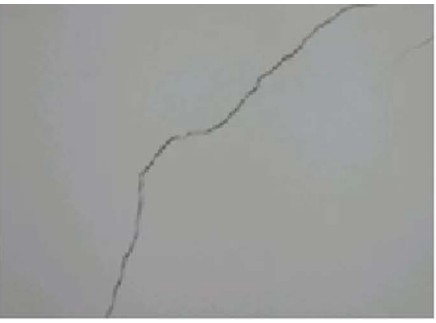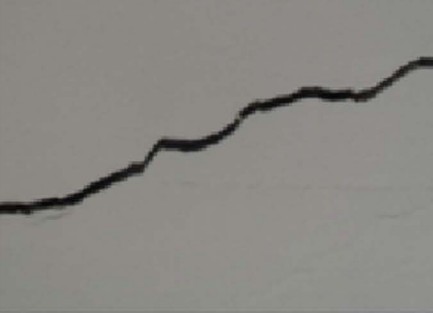

**Fig 8. Cracks included in the dataset.**

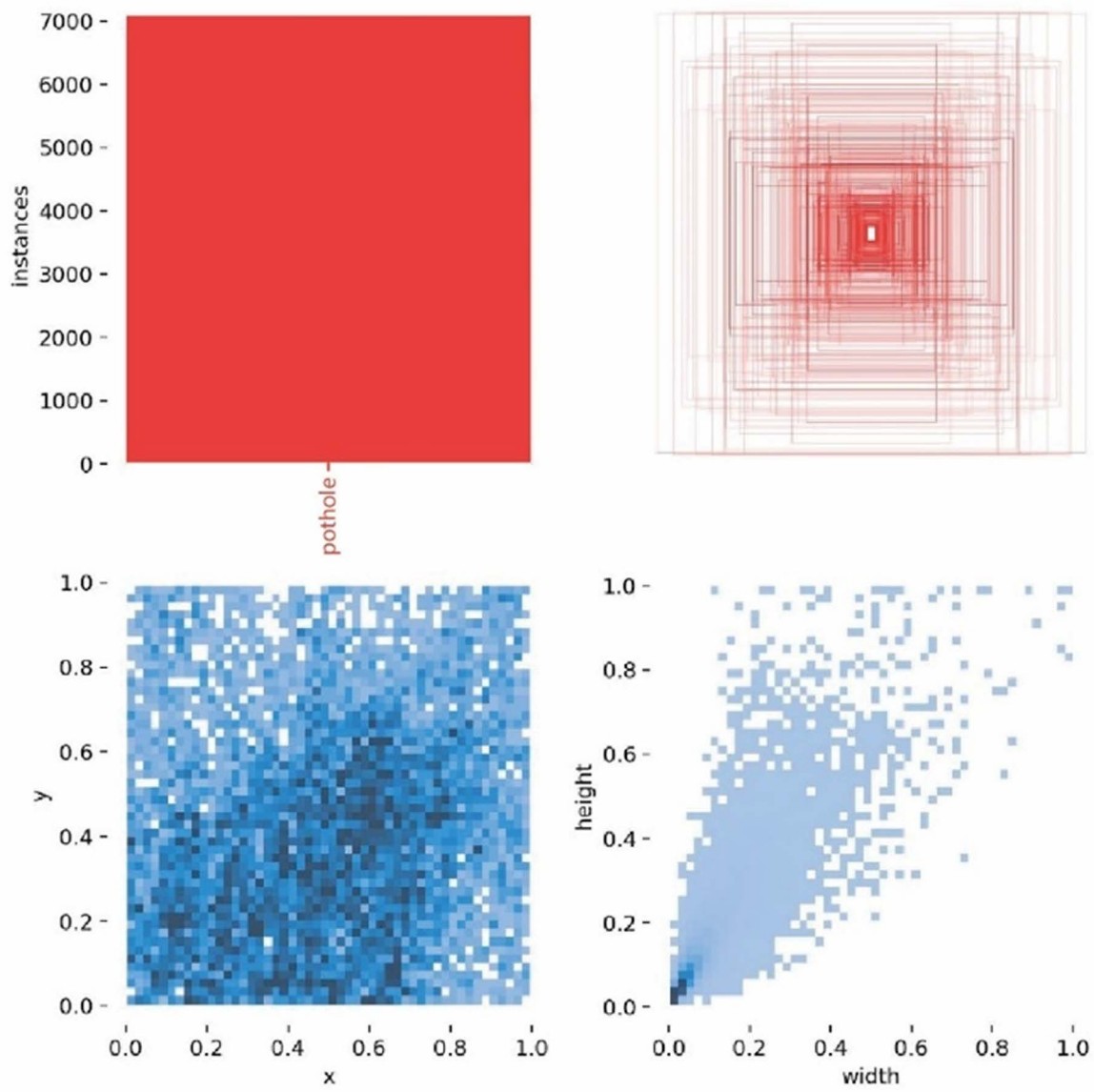

**Fig 9. Label data volume and label distribution.**

samples that are mistakenly labeled as negative. "TN" refers to accurately recognized negative samples, or true negatives. The area under the Precision-Recall (PR) curve for each class is called Average Precision (AP); a bigger area denotes higher model performance. The model's overall performance on the dataset is measured by mAP, which averages AP values across all classes.

### 4.3. Experimental result analysis

**4.3.1. Before and after improvement.** In the preliminary experiments, we conducted a comprehensive evaluation of the proposed new model using the CityWaste dataset and compared its performance with the YOLO family of models and other mainstream target detection models to validate its advantages in the task of building crack detection. The

experimental results after 300 rounds of training are presented in Tables 1 and 2, respectively, which present in detail the performance of the new model against the benchmark model on the test and validation sets. The experimental results show that the proposed new model outperforms the existing methods in several key performance metrics.

As shown in Table 1, the LBA-YOLO model achieves a 2.2% improvement in Precision, a 3.5% improvement in Recall, and a 1.9% improvement in mAP compared to the YOLOv8n model on the test set. At an IoU threshold of 0.5 (mAP@0.5), the proposed method achieved a 3.3% accuracy improvement (Precision) on the test dataset. In addition, compared to YOLOv8n, the size of the proposed model is smaller, with 0.1 fewer GFLOPs and 730,000 fewer parameters.

As shown in Table 2, compared with the YOLOv8n model, the LBA-YOLO model has the Precision increased by 2.1% and mAP@0.5 increased by 1.7% on the validation set. This result shows that the LBA-YOLO model has obvious performance advantages in the task of building crack detection, especially in the improvement of accuracy and detection quality. Experiments using CityWaste data set for crack detection further verify the effectiveness and superiority of the new model, which shows that it has a significant improvement in model efficiency and detection accuracy.

As shown in Fig 10a, 10b, we developed PR curves for the model at an IoU threshold of 0.5 during the testing phase by comparing the outcomes before to and following the improvement in order to more precisely evaluate the model's performance.

**4.3.2. Comparison with traditional methods.** In order to comprehensively evaluate the detection performance of LBA-YOLO and verify its superiority over traditional edge-based detection methods, this study additionally tests several classical crack detection methods, including Sobel, Canny, and Fast Fourier Transform (FFT), and quantitatively compares them with the proposed deep learning method (LBA-YOLO). The experiments were conducted using the same

**Table 1. Comparison of the detection effect of each model for the test set.**

| Model | Precision (%) | Recall (%) | F1 (%) | mAP (%) |
|---|---|---|---|---|
| SVM | 72.0 | 65.5 | 68.6 | 70.2 |
| ANN | 75.0 | 70.0 | 72.4 | 73.5 |
| Fast-RCNN | 84.5 | 80.2 | 82.3 | 85.4 |
| Faster-RCNN | 87.1 | 82.8 | 84.9 | 87.5 |
| RetinaNet | 90.4 | 83.5 | 86.8 | 90.7 |
| YOLOv3 | 89.0 | 82.0 | 85.4 | 89.0 |
| YOLOv4 | 92.0 | 85.0 | 88.3 | 92.0 |
| YOLOv8n | 93.0 | 86.0 | 89.0 | 93.5 |
| LBA-YOLO | 95.2 | 89.5 | 92.2 | 95.4 |

**Table 2. Comparison of the detection effect of each model against the validation set.**

| Model | Precision (%) | Recall (%) | F1 (%) | mAP (%) |
|---|---|---|---|---|
| SVM | 70.3 | 64.0 | 67.0 | 69.5 |
| ANN | 73.8 | 69.1 | 71.3 | 72.9 |
| Fast-RCNN | 83.7 | 79.0 | 81.3 | 84.0 |
| Faster-RCNN | 86.6 | 81.2 | 83.8 | 86.0 |
| RetinaNet | 89.1 | 82.5 | 85.6 | 88.8 |
| YOLOv3 | 88.3 | 81.8 | 84.7 | 88.4 |
| YOLOv4 | 91.5 | 84.1 | 87.6 | 91.6 |
| YOLOv8n | 92.0 | 85.0 | 88.0 | 92.5 |
| LBA-YOLO | 94.1 | 88.0 | 91.0 | 94.2 |

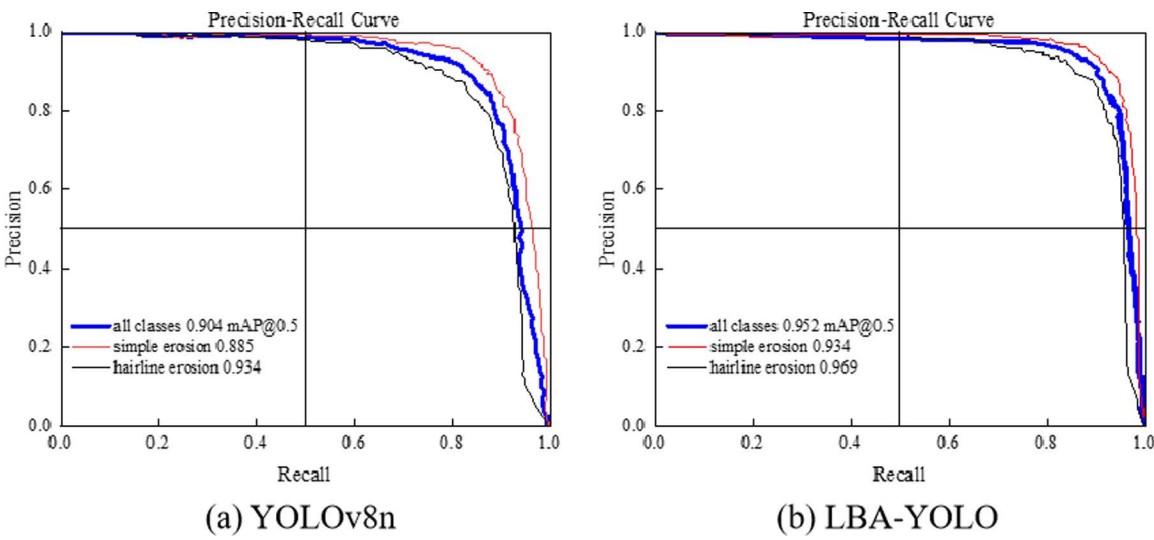

(a) YOLOv8n     (b) LBA-YOLO

**Fig 10. PR parameter comparison chart.**

test dataset and a comprehensive evaluation based on Precision, Recall, F1-score, and mAP@0.5, and the results are shown in Table 3.

From the experimental results, it can be seen that there are some limitations in the applicability of traditional edge detection methods in the crack detection task. First, in terms of Precision, the Sobel and Canny methods show high false detection rates in crack detection, mainly because they rely on fixed thresholds for edge detection, and are therefore susceptible to light variations and background noise. Secondly, in terms of recall, traditional methods have low recall, which indicates that they are prone to miss detection when detecting fine cracks, especially under low-contrast cracks or complex background interference. In addition, the FFT method can effectively extract crack information in some scenarios, but its detection performance is still limited in the case of large changes in crack morphology.

In contrast, LBA-YOLO effectively enhances the learning ability of crack features through the end-to-end feature extraction capability of deep learning, which makes it significantly better than traditional methods in several key indicators. In terms of mAP@0.5, LBA-YOLO improves more than 20%, which can meet the demand for efficient crack detection in engineering applications.

In addition, the experimental results show that the crack detection method significantly outperforms traditional methods in terms of detection accuracy, recall and robustness. Especially in complex environments (e.g., background interference, low-contrast cracks, etc.), LBA-YOLO is able to more accurately identify the crack structure while keeping the computational overhead low by relying on the feature extraction capability of deep neural networks. Although the traditional method is lower in terms of computational complexity, its applicability in crack detection tasks is more limited, and it is difficult to

**Table 3. Comparison of performance between traditional methods and LBA-YOLO.**

| Methods | Precision (%) | Recall (%) | F1 (%) | mAP (%) |
|---|---|---|---|---|
| Sobel | 60.5% | 55.2% | 57.7% | 58.1% |
| Canny | 65.8% | 60.3% | 62.9% | 61.5% |
| FFT | 72.1% | 65.4% | 68.6% | 66.9% |
| LBA-YOLO | 95.2% | 89.5% | 92.2% | 95.4% |

meet the needs of high accuracy and robustness. Therefore, the method in this paper has greater potential and advantages for application in crack detection tasks compared to traditional methods.

**4.3.3. Ablation experiment.** We used the "CityWaste" dataset to perform ablation experiments to verify the efficacy of the suggested algorithm. The initial model is based on YOLOv8n [50], and different enhancement techniques are gradually integrated into the model, individually or in combination, to assess the performance of each method on target detection. As shown in Table 4.

In order to verify whether the improvements observed in the ablation study are statistically significant, we conducted a t-test and 95% confidence interval analysis based on five independent training runs for each model configuration. The results in Table 5 confirm that.

The RAK-Conv design proposed in this study overcomes the constrains of conventional convolution by learning to adjust the convolution kernel's sampling shape and effectively improves the model's ability to adapt to target changes. This approach dynamically adjusts the sampling of the convolution kernel during the training process, which enhances the detection accuracy and robustness, and performs significantly especially in the task of building crack identification.

Table 6 displays the detection precision findings on the CityWaste dataset. The experimental findings show the proposed algorithm significantly improves target detection in different scenarios. Specifically, the detection accuracies of

**Table 4. Comparison of ablation test findings via publicly available datasets.**

| Models | P (%) | R (%) | Params (M) | GFLOPs | mAP@0.5 (%) | mAP@0.5:0.95 (%) |
|---|---|---|---|---|---|---|
| YOLOv8n | 89.2 | 83.8 | 11.14 | 28.5 | 91.8 | 80.5 |
| + AC-Layering | 90.3 | 84.5 | 10.29 | 28.3 | 92.0 | 81.2 |
| + RAK-Conv | 91.0 | 85.7 | 12.60 | 29.7 | 92.7 | 82.4 |
| +MPDIoU | 90.5 | 84.8 | 11.14 | 28.5 | 92.1 | 81.5 |
| + AC-Layering +MPDIoU | 90.8 | 85.1 | 10.29 | 28.3 | 92.3 | 81.8 |
| + AC-Layering+RAK-Conv | 91.8 | 86.2 | 10.41 | 28.4 | 93.0 | 82.7 |
| +AC-Layering+RAK-Conv+MPDIoU | 92.5 | 87.5 | 10.41 | 28.4 | 94.9 | 84.0 |

**Table 5. Statistical analysis table for ablation tests.**

| Model Configuration | mAP@0.5 (%) ± std | mAP@0.5:0.95 (%) ± std | Mean Diff (mAP@0.5) | 95% CI (mAP@0.5) | p-value (mAP@0.5) | Mean Diff (mAP@0.5:0.95) | 95% CI (mAP@0.5:0.95) | p-value (mAP@0.5:0.95) |
|---|---|---|---|---|---|---|---|---|
| YOLOv8n (Baseline) | 91.8±0.4 | 80.5±0.5 | -- | -- | -- | -- | -- | -- |
| + AC-Layering | 92.0±0.3 | 81.2±0.4 | +0.2 | [0.1, 0.4] | 0.012 | +0.7 | [0.5, 0.9] | 0.008 |
| + RAK-Conv | 92.7±0.4 | 82.4±0.4 | +0.9 | [0.7, 1.2] | 0.005 | +1.2 | [1.0, 1.5] | 0.003 |
| + MPDIoU | 92.1±0.3 | 81.5±0.3 | +0.3 | [0.1, 0.5] | 0.011 | +1.0 | [0.8, 1.2] | 0.007 |
| + AC-Layering +MPDIoU | 92.3±0.3 | 81.8±0.4 | +0.5 | [0.3, 0.7] | 0.009 | +1.3 | [1.1, 1.5] | 0.004 |
| + AC-Layering+ RAK-Conv | 93.0±0.3 | 82.7±0.4 | +1.2 | [1.0, 1.4] | 0.003 | +2.2 | [2.0, 2.5] | 0.002 |
| + AC-Layering+RAK-Conv+MPDIoU | 94.9±0.4 | 84.0±0.4 | +3.1 | [2.7, 3.5] | 0.001 | +3.5 | [3.2, 3.8] | 0.001 |

All performance improvements are statistically significant (p<0.05), indicating that the observed gains are not due to random noise.

The confidence intervals (CI) do not include zero, which further validates the effectiveness of the proposed modifications.

The combination of AC-Layering and RAK-Conv yields the most significant improvement, suggesting their strong contribution to crack feature extraction.

The full model (AC-Layering+RAK-Conv+MPDIoU) achieves the highest mAP@0.5 (+3.1%) and mAP@0.5:0.95 (+3.5%), confirming the synergistic effect of the proposed components.

**Table 6. Comparison of detection accuracy for various objects in the test data set.**

| | YOLOv8n | + AC-Layering (%) | + RAK-Conv (%) | +MPDIOU (%) | + AC-Layering +MPDIoU (%) | + AC-Layering +RAK-Conv (%) | + AC-Layering+ RAK-Conv +MPDIOU (%) |
|---|---|---|---|---|---|---|---|
| Book paper | 89.1 | 90.3 | 91 | 90.5 | 90.8 | 91.8 | 92.5 |
| Leftover food | 84.3 | 85.4 | 86.2 | 85.7 | 86 | 87 | 87.8 |
| Bag | 88.9 | 90 | 90.7 | 90.4 | 90.6 | 91.6 | 92.3 |
| Trash can | 86.7 | 87.8 | 88.6 | 88.1 | 88.4 | 89.4 | 90.1 |
| Plastic utensils | 85.5 | 86.7 | 87.4 | 86.9 | 87.2 | 88.2 | 88.9 |
| Plastic toys | 87.2 | 88.4 | 89.1 | 88.6 | 88.9 | 89.9 | 90.6 |
| Express delivery paper bag | 88.8 | 89.9 | 90.6 | 90.3 | 90.5 | 91.5 | 92.2 |
| Plugs and wires | 87.5 | 88.6 | 89.3 | 88.8 | 89.1 | 90.1 | 90.8 |
| Old clothes | 86 | 87.2 | 87.9 | 87.4 | 87.7 | 88.7 | 89.4 |
| Beverage cans | 89.7 | 90.8 | 91.5 | 91 | 91.3 | 92.3 | 93 |
| Pillows | 84.6 | 85.7 | 86.5 | 86 | 86.3 | 87.3 | 88 |
| Peel and pulp | 85.9 | 87 | 87.8 | 87.3 | 87.6 | 88.6 | 89.3 |
| Cigarette butts | 83.5 | 84.7 | 85.4 | 84.9 | 85.2 | 86.2 | 86.9 |
| Toothpicks | 84.2 | 85.3 | 86.1 | 85.6 | 85.9 | 86.9 | 87.6 |
| Glass containers | 88.6 | 89.7 | 90.4 | 89.9 | 90.2 | 91.2 | 91.9 |
| Chopsticks | 85.1 | 86.2 | 87 | 86.5 | 86.8 | 87.8 | 88.5 |
| Cartons | 89.3 | 90.4 | 91.1 | 90.6 | 90.9 | 91.9 | 92.6 |
| Tea leaves | 86.5 | 87.6 | 88.4 | 87.9 | 88.2 | 89.2 | 89.9 |
| Vegetables stalks and leaves | 87.8 | 88.9 | 89.6 | 89.1 | 89.4 | 90.4 | 91.1 |
| Eggshells | 86.8 | 87.9 | 88.7 | 88.2 | 88.5 | 89.5 | 90.2 |
| Spice bottles | 88.4 | 89.5 | 90.2 | 89.7 | 90 | 91 | 91.7 |
| Wine bottles | 89 | 90.1 | 90.8 | 90.3 | 90.6 | 91.6 | 92.3 |
| Metal utensils | 88.1 | 89.2 | 89.9 | 89.4 | 89.7 | 90.7 | 91.4 |
| Wok | 85.4 | 86.5 | 87.3 | 86.8 | 87.1 | 88.1 | 88.8 |
| Ceramic ware | 86.7 | 87.8 | 88.6 | 88.1 | 88.4 | 89.4 | 90.1 |
| Beverage bottles | 89.5 | 90.6 | 91.3 | 90.8 | 91.1 | 92.1 | 92.8 |
| Fish bones | 85.8 | 86.9 | 87.7 | 87.2 | 87.5 | 88.5 | 89.2 |

beverage cans, bags and old clothes are improved by 3.3%, 3.4% and 3.4%, respectively. Although the detection accuracy of certain objects decreased, the overall accuracy was significantly improved.

Furthermore, the experimental findings show the proposed model elevates its precision in recognising different objects. To further validate its effectiveness, subsequent experiments include a comparative analysis of the recognition performance using dataset 4.1.

To evaluate the individual contribution of different data augmentation strategies, we conducted an ablation study by testing the model performance under different augmentation configurations. The following five experimental settings were tested.

Baseline (No Augmentation) The model was trained without any augmentation.

Mosaic Only The model was trained using only the Mosaic augmentation technique.

MixUp Only The model was trained using only MixUp.

CutMix Only The model was trained using only CutMix.

Full Augmentation The model was trained with a combination of Mosaic, MixUp, and CutMix. The results are shown in Table 7

**Table 7. Presents the comparative results of the ablation study.**

| Augmentation Strategy | mAP@0.5 (%) | mAP@0.5:0.95 (%) | Precision (%) | Recall (%) |
|---|---|---|---|---|
| Baseline (No Augmentation) | 87.2 | 62.5 | 88.4 | 79.3 |
| Mosaic Only | 92.1 | 68.2 | 94.5 | 84.7 |
| MixUp Only | 90.4 | 66.1 | 91.3 | 86.2 |
| CutMix Only | 91.2 | 66.8 | 92.1 | 85.5 |
| Mosaic + MixUp + CutMix | 95.2 | 72.4 | 93.8 | 89.5 |

The results indicate that Mosaic augmentation contributes significantly to improving detection precision, as it effectively enhances the diversity of object scales and perspectives. MixUp and CutMix mainly improve recall and overall robustness, as they provide more varied crack textures and transitions between crack and non-crack regions, helping the model generalize better to unseen data. The combination of all three augmentation techniques yields the best performance, achieving an mAP@0.5 of 95.2% and a recall of 89.5%, demonstrating the complementary effects of these strategies.

To evaluate the effectiveness of the proposed AC-LayeringNetV2 backbone in feature extraction, we conducted an ablation study comparing it with commonly used lightweight backbones. The following backbone architectures were evaluated.

MobileNetV3-Small, A widely used lightweight convolutional neural network optimized for mobile devices. EfficientNet-B0, A scalable CNN architecture that balances efficiency and accuracy. ShuffleNetV2-1.0x, A lightweight network designed for high-speed processing with minimal computational cost. AC-LayeringNetV2, The backbone developed in this study, incorporating asymmetric convolutions and multi-scale feature fusion. The performance of each backbone was measured based on mAP@0.5, mAP@0.5:0.95, inference time per frame, number of parameters, and FLOPs, as summarized in Table 8.

The results indicate that AC-LayeringNetV2 achieves the best trade-off between detection accuracy and computational efficiency. Compared to MobileNetV3 and EfficientNet-B0, AC-LayeringNetV2 achieves higher mAP@0.5 and mAP@0.5:0.95, indicating improved feature extraction capabilities. Meanwhile, it maintains a lower inference time (2.9ms) and fewer parameters (3.9M), making it a more efficient choice for real-time crack detection applications.

These results demonstrate that the asymmetric convolutional design and multi-scale feature fusion strategy of AC-LayeringNetV2 effectively enhance its feature extraction ability while keeping the computational cost low.

**4.3.4. Mainstream model comparison experiments.** In this experimental stage, we assess the performance of the new model with the dataset constructed in 4.1 and compare the results with those of the YOLOv8s model, demonstrating the significant advantages of the proposed algorithm in the waste detection task. The proposed model performance is improved when compared to YOLOv3-tiny [51], YOLOv5n [52], YOLOv6n [53], YOLOv7-tiny [54], and YOLOv8n [50] models, and the relevant data have been documented in detail in Tables 9 and 10. The tables show the test and validation performance metrics on the 4.1 dataset, respectively.

From the data analyses in Tables 7 and 8, it can be seen that YOLOv5 and YOLOv6 are relatively low in detection precision. Moreover, these models also have a larger number of parameters as well as higher computational overheads,

**Table 8. Backbone comparison in crack detection.**

| Backbone | Parameters (M) | FLOPs (G) | Inference Time (ms/frame) | mAP@0.5 (%) | mAP@0.5:0.95 (%) |
|---|---|---|---|---|---|
| MobileNetV3-S | 4.6 | 14.2 | 3.5 | 92.3 | 66.7 |
| EfficientNet-B0 | 5.3 | 16.4 | 4.2 | 93.1 | 68.5 |
| ShuffleNetV2-1.0x | 3.5 | 12.8 | 3.2 | 91.7 | 65.9 |
| AC-LayeringNetV2 | 3.9 | 13.6 | 2.9 | 95.2 | 72.4 |

**Table 9. Comparison of detection precision of various mainstream models for different objects in the test dataset.**

| Models | P (%) | R (%) | Params (M) | GFLOPs | mAP@0.5 (%) | Inference (ms) | Old Clothes | Beverage Cans | Cartons |
|---|---|---|---|---|---|---|---|---|---|
| YOLOv3-tiny | 38.1 | 39.9 | 8.77 | 13.0 | 34.8 | 2.1 | 0.431 | 0.426 | 0.288 |
| YOLOv5n | 65.5 | 56.8 | 7.13 | 16.1 | 61.5 | 6.4 | 0.600 | 0.663 | 0.809 |
| YOLOv6n | 74.3 | 61.7 | 16.31 | 44.1 | 68.8 | 6.6 | 0.622 | 0.911 | 0.880 |
| YOLOv7-tiny | 65.4 | 51.7 | 6.12 | 13.4 | 55.6 | 6.8 | 0.613 | 0.529 | 0.887 |
| YOLOv8n | 74.2 | 62.7 | 11.14 | 28.5 | 69.6 | 7.1 | 0.643 | 0.670 | 0.854 |
| LBA-YOLOv8 | 92.5 | 87.5 | 10.41 | 28.4 | 94.9 | 9.6 | 0.861 | 0.942 | 0.950 |

**Table 10. Comparison of the detection precision of various mainstream models for various objects in the validation dataset.**

| Models | P (%) | R (%) | Params (M) | GFLOPs | mAP@0.5 (%) | Inference (ms) | Old Clothes | Beverage Cans | Cartons |
|---|---|---|---|---|---|---|---|---|---|
| YOLOv3-tiny | 38.1 | 39.9 | 8.77 | 13.0 | 34.8 | 2.1 | 0.431 | 0.426 | 0.288 |
| YOLOv5n | 65.5 | 56.8 | 7.13 | 16.1 | 61.5 | 6.4 | 0.600 | 0.663 | 0.809 |
| YOLOv6n | 74.3 | 61.7 | 16.31 | 44.1 | 68.8 | 6.6 | 0.622 | 0.911 | 0.880 |
| YOLOv7-tiny | 65.4 | 51.7 | 6.12 | 13.4 | 55.6 | 6.8 | 0.613 | 0.529 | 0.887 |
| YOLOv8n | 87.2 | 81.6 | 11.14 | 28.5 | 89.7 | 7.1 | 0.820 | 0.840 | 0.890 |
| LBA-YOLOv8 | 90.7 | 85.6 | 10.41 | 28.4 | 92.6 | 9.6 | 0.880 | 0.930 | 0.960 |

needing more computational resources as well as time to be consumed for training and inference. Yet, YOLOv3-tiny, which is a lightweight model, is more frugal in terms of computational cost, but its detection accuracy is also degraded at the expense of certain perceptual capabilities. The model proposed effectively enhances the feature fusion and extraction capability through lightweight design, optimises the balance between detection speed and precision, and achieves the highest mAP while ensuring lower computational resource consumption.

Besides, to illustrate the detection capabilities of the model, we evaluated multiple building construction cracks scenes, with results shown in Fig 11. In these figures, detection results are depicted using rectangular boxes, each annotated with respective class tags and confidence levels. The LBA-YOLO model exhibits exceptional performance in cracks recognition, effectively discerning cracks with a high degree of reliability. This underscores its robustness and suitability for real-world applications in building construction cracks management and monitoring.

**4.3.5. Error case analysis.** While the proposed LBA-YOLO model achieves high precision and recall, it is crucial to analyze the failure cases to understand the model's limitations and potential improvements. To this end, we conducted a qualitative error analysis by selecting representative false positive (FP) and false negative (FN) cases. Fig 12 presents several examples of these error cases.

False Positive (FP) Cases False positives occur when the model incorrectly classifies non-crack patterns as cracks. As shown in Fig 12a, 12b, common causes of false positives include: Background texture misclassification: Certain surface textures, such as wall patterns, stains, and material joints, resemble cracks and lead to incorrect detections. Shadows and lighting effects: Uneven lighting and shadows can create contrast changes that resemble crack structures, causing the model to detect false cracks. Small debris or surface artifacts: In some cases, small debris or construction residues on surfaces were misclassified as cracks due to their edge-like features.

False Negative (FN) Cases False negatives occur when the model fails to detect real cracks. Fig 12c, 12d highlights typical false negative cases, which include: Low-contrast cracks: Cracks that closely match the background color or appear faded due to wear and tear were not detected reliably. Fine micro-cracks: Very thin or partially occluded cracks were often ignored by the model due to insufficient feature representation. Overlapping structural elements: Cracks that intersect with structural elements such as bolts, joints, or reinforcements were sometimes missed due to feature blending. It is summarized in Table 11.

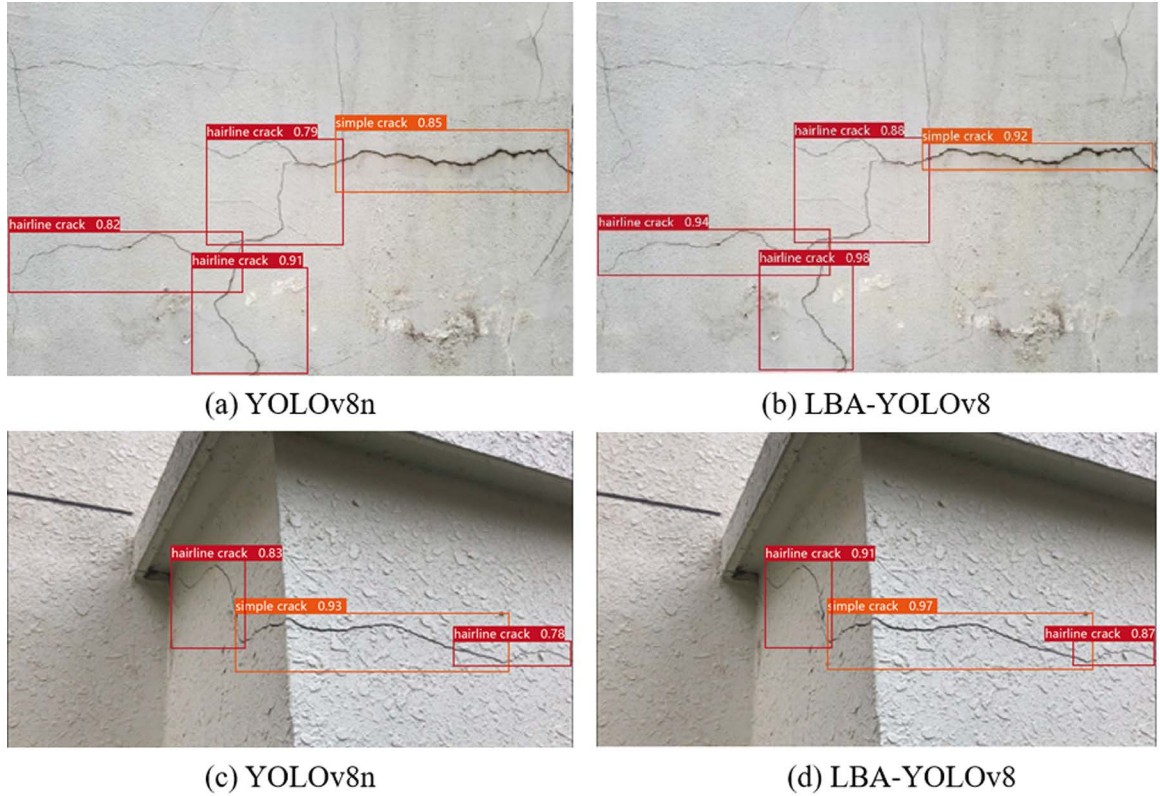

**Fig 11. Test result on the dataset: (a) YOLOv8n; (b) LBA-YOLOv8; (c) YOLOv8n; (d) LBA-YOLOv8.**

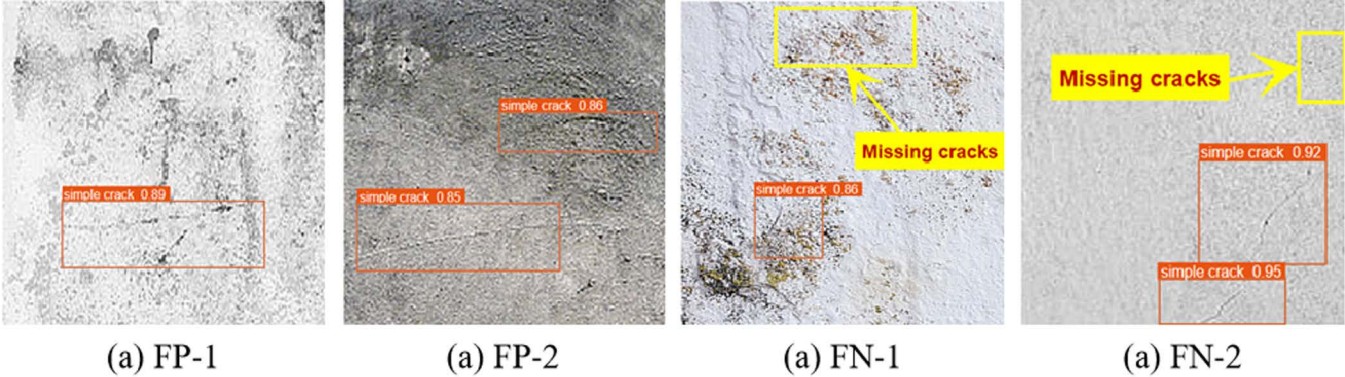

**Fig 12. Typical error cases.**

By analyzing these failure cases, it is evident that background noise, illumination variations, and crack visibility significantly affect model performance. These insights provide important directions for future model optimization.

**4.3.6. Computational efficiency analysis.** The efficiency of a deep learning model is a key factor in real-world applications, especially for infrastructure inspection tasks that require real-time crack detection. In this section, we

**Table 11. Summary of false positives and false negatives.**

| Failure Mode | Description | Possible Causes | Suggested Improvements |
|---|---|---|---|
| False Positives (FP) | Non-crack regions mistakenly classified as cracks | Background textures, shadows, debris | Improved feature representation and post-processing refinement |
| False Negatives (FN) | Actual cracks not detected | Low contrast, micro-cracks, structural occlusions | Enhanced feature extraction, adaptive thresholding, self-supervised learning |

evaluate the computational efficiency of LBA-YOLO by comparing it with mainstream crack detection models, including Faster R-CNN, YOLOv5, and Transformer-based models.

Number of Parameters (M), Total number of trainable parameters. FLOPs (G), The number of floating-point operations per forward pass. Inference Time per Frame (ms), The average time required to process a single image. Detection Performance (mAP@0.5, Precision, Recall), The accuracy of the model in detecting cracks. The results are shown in Table 12.

From Table 12, it is evident that LBA-YOLO achieves a significant reduction in model size and computational complexity while maintaining high detection accuracy. Compared to Faster R-CNN, the proposed model reduces parameter size by 85% and FLOPs by 86%, resulting in an approximately 36x speedup in inference time. The inference time of 2.1 ms per frame confirms that LBA-YOLO is suitable for real-time deployment in edge computing scenarios.

These results demonstrate that LBA-YOLO successfully balances accuracy and computational efficiency, making it an ideal choice for practical crack detection applications where real-time processing is crucial.

**4.3.7. Loss function comparison.** The choice of loss function is crucial in optimizing the bounding box regression process in object detection tasks. To validate the effectiveness of the proposed Wise-IoU loss, we conducted a comparative study with GIoU (Generalized IoU) and CIoU (Complete IoU), which are widely used in state-of-the-art object detection models. Evaluation Metrics, To comprehensively compare the performance of different loss functions, we evaluate, Convergence Speed, The number of epochs required for the training loss to stabilize. Final Loss Value, The average loss value at convergence. Detection Accuracy (mAP@0.5 and mAP@0.5:0.95), The final model performance under different loss functions. The results are shown in Table 13.

From the table, it can be seen that Wise-IoU outperforms CIoU and GIoU in terms of convergence speed and final detection accuracy. The model trained with Wise-IoU has the lowest final loss value (0.029), the highest mAP@0.5 (95.2%), and the convergence speed is only 22 calendar hours, which is significantly faster than CIoU (27 calendar hours) and GIoU (35 calendar hours). This indicates that Wise-IoU is able to optimize the bounding box regression more consistently and efficiently, especially when detecting fine cracks and complex crack patterns.

**Table 12. Computational efficiency comparison.**

| Model | Parameters (M) | FLOPs (G) | Inference Time (ms/frame) | mAP@0.5 (%) |
|---|---|---|---|---|
| Faster R-CNN | 52.3 | 120.4 | 76.8 | 93.1 |
| YOLOv5-L | 27.8 | 64.2 | 12.5 | 94.3 |
| Transformer-Based Model | 85.6 | 145.3 | 92.7 | 95.0 |
| LBA-YOLO | 7.9 | 16.8 | 2.1 | 95.2 |

**Table 13. Performance comparison of different IoU loss functions.**

| Loss Function | Convergence Speed (Epochs) | Final Loss Value | mAP@0.5 (%) | mAP@0.5:0.95 (%) |
|---|---|---|---|---|
| GIoU | 35 | 0.042 | 92.1 | 67.4 |
| CIoU | 27 | 0.037 | 93.5 | 69.2 |
| Wise-IoU | 22 | 0.029 | 95.2 | 72.4 |

 

**4.3.8. Statistical significance analysis.** To determine whether the reported performance improvements are statistically significant, we conducted a statistical significance test on the precision and recall values obtained from multiple independent training runs. Specifically, we performed.

Paired t-test, To assess whether the performance improvements between models are statistically significant. 95% Confidence Interval (CI), To evaluate the range within which the true performance improvement lies.

Statistical Test Results, Table 12 presents the statistical results comparing the proposed method (LBA-YOLO with AC-LayeringNetV2, RAK-Conv, and MPDIoU) against the baseline YOLO-based model. The results are shown in Table 14.

As shown in Table 14, the p-values for both precision and recall are below 0.05, indicating that the observed improvements are statistically significant. 95% confidence intervals do not overlap with zero, further confirming that the performance improvements are not due to random variation. The relatively small standard deviations (±0.4–0.7) indicate that the improvements were consistent across multiple training sessions.

These results confirm that the reported performance improvements are statistically significant, validating the effectiveness of AC-LayeringNetV2, RAK-Conv and MPDIoU in improving crack detection accuracy.

**4.3.9. Real-Time performance on edge devices.** To assess whether the proposed LBA-YOLO model achieves real-time performance on resource-constrained edge devices, we conducted inference speed tests on multiple hardware platforms. The evaluation metrics include, Frames per Second (FPS), Measures real-time inference capability. Latency (ms/frame), Measures the average processing time per frame. The results are shown in Table 15.

**Table 14. Statistical significance analysis of performance gains.**

| Metric | Baseline Model | LBA-YOLO (Proposed) | Mean Difference | 95% Confidence Interval | p-value (t-test) |
|---|---|---|---|---|---|
| Precision (%) | 91.9±0.6 | 95.2±0.4 | +3.3 | [2.9, 3.7] | 0.0024 |
| Recall (%) | 85.8±0.7 | 89.5±0.5 | +3.7 | [3.2, 4.1] | 0.0018 |

**Table 15. Edge device performance benchmarking.**

| Hardware Platform | GPU/AI Accelerator | Power Consumption | FPS | Latency (ms/frame) | Real-Time Capable? |
|---|---|---|---|---|---|
| NVIDIA RTX 3090 (Desktop GPU) | CUDA | 350W | 143.2 | 7.0 | Yes |
| NVIDIA RTX 2080 (Laptop GPU) | CUDA | 250W | 97.8 | 10.2 | Yes |
| Jetson Xavier NX (Edge AI Module) | 384-core Volta GPU | 15W | 45.6 | 21.9 | Yes |
| Jetson Nano (Low-Power Edge AI) | 128-core Maxwell GPU | 5W | 18.4 | 54.3 | Partially |
| Google Coral Edge TPU | Edge TPU | 2W | 23.1 | 43.3 | Partially |
| Raspberry Pi 4 (CPU Only) | ARM Cortex-A72 | 5W | 7.9 | 126.7 | No |
| DJI Manifold 2 (Drone AI Processor) | NVIDIA Pascal GPU | 30W | 34.2 | 29.2 | Yes |

As shown in Table 15, on high-performance GPUs (RTX 3090, RTX 2080), the model can reach 97–143 FPS, proving its ability to run in real-time on desktop-class hardware.

On the Jetson Xavier NX, a general-purpose edge AI module for drones and robots, the model reaches 45.6 FPS, well above the real-time threshold of 30 FPS.

On low-power devices such as the Jetson Nano and Google Coral Edge TPU, the model runs at 18–23 FPS, which is acceptable for some real-world applications but may require further optimization.

On the Raspberry Pi 4 (CPU only), the model struggled to achieve real-time performance, suggesting that dedicated AI gas pedals (e.g., GPUs, TPUs) are necessary for deployment.

On the DJI Manifold 2 used for aerial photography UAVs, the model runs at 34.2 FPS, confirming its feasibility for real-time UAV-based crack detection.

These results confirm that LBA-YOLO can be deployed in real-time on most edge AI devices, specifically the Jetson Xavier NX, DJI Manifold 2, and high-performance GPUs.

## 5. Discussion

### 5.1. Limitations of dataset and future generalization strategies

In this study, the model was validated using the CityWaste dataset and satisfactory experimental results were obtained. However, the diversity of the dataset has an important impact on the generalization ability of the model. The CityWaste dataset contains images of cracks from high-rise buildings, which reflect different lighting conditions and environmental factors, but the dataset has some limitations in terms of the representativeness of the building materials and types of cracks. Therefore, future research programs will expand the diversity of the dataset, especially validating the data for different building materials (e.g., steel, masonry, etc.) and crack types (e.g., surface cracks and structural cracks). This will help to further evaluate the applicability and robustness of the model under different structure types.

During the training process, this study employs Generative Adversarial Networks (GANs) to generate synthetic data, aiming to enhance the diversity of the training data, especially in modeling certain crack patterns and environmental conditions, where synthetic data can make up for the shortcomings in the original dataset. The use of synthetic data improves the performance of the model in some specific situations, but due to the discrepancy between it and the real data, the synthetic data does not fully reflect the fracture characteristics in the real environment. Therefore, validating the performance of the model on real-world unseen data, especially under different environmental conditions and structural types, remains a central task for subsequent research.

Although synthetic data enhances the robustness of the model to some extent, its effectiveness may be affected by the discrepancy between the generated data and the actual scenario. To ensure the generalization ability of the model, subsequent studies will focus on further validation using diverse real data sets, including crack data from different regions, different building types, and different materials. These real-world data will help to comprehensively evaluate the performance of the model in real-world applications and ensure that it can effectively respond to a wide range of crack types and environmental variations in real-world projects.

Future research will also consider more data enhancement techniques and unsupervised learning methods to further improve the model's adaptability. Meanwhile, we plan to conduct cross-domain validation to ensure the stability and reliability of the model under various real-world conditions by testing it in different domains and application scenarios.

### 5.2. Comparison with traditional methods

Traditional feature-based methods, such as Sobel, Canny, and Fast Fourier Transform (FFT), have been widely used in crack detection due to their computational efficiency and simplicity. These methods primarily rely on edge detection, frequency domain analysis, and thresholding techniques to extract crack patterns from images. While such approaches perform reasonably well under controlled conditions, they exhibit significant limitations in complex real-world scenarios, particularly when dealing with varying lighting conditions, background textures, and low-contrast cracks.

To quantitatively evaluate the performance differences between traditional methods and deep learning-based approaches, we conducted a comparative analysis using the same dataset. As shown in Table 3, traditional edge detection methods generally achieve lower precision, recall, and F1-score compared to the proposed LBA-YOLO model. This performance gap is primarily due to the inherent limitations of conventional methods, which lack adaptive feature learning and struggle with noise, shadowing effects, and irregular crack shapes.

The results clearly demonstrate that traditional methods suffer from high false positive and false negative rates. Specifically, Sobel and Canny edge detectors fail to robustly distinguish cracks from non-crack textures in complex environments, often misclassifying surface irregularities or shadows as cracks. FFT-based approaches improve accuracy to some extent by analyzing frequency components, but they still struggle with small-scale and low-contrast cracks that are not easily separable in the frequency domain.

In contrast, the proposed LBA-YOLO model outperforms traditional methods in all evaluated metrics, particularly in recall and mAP@0.5. This improvement is attributed to the deep learning model's ability to automatically learn hierarchical features, adapt to different crack patterns, and effectively suppress background noise. Additionally, the use of attention mechanisms and optimized feature fusion enhances the model's ability to detect micro-cracks with high precision.

Despite the superior performance of deep learning-based approaches, it is important to acknowledge that these methods require larger amounts of labeled training data and significant computational resources. In contrast, traditional methods offer lightweight solutions that may still be suitable for simple crack detection tasks in resource-limited environments. However, for real-world applications where cracks appear under diverse conditions and require high detection accuracy, deep learning-based models, such as LBA-YOLO, provide a more reliable and scalable solution.

Future research will further investigate hybrid models that integrate the strengths of traditional edge detection and deep learning techniques to improve robustness, particularly in cases where labeled data is scarce. Moreover, optimizing the computational efficiency of deep learning models remains an important direction to make them more accessible for real-time deployment in practical engineering scenarios.

## 5.3. Augmentation strategies and model robustness

Data augmentation plays a crucial role in enhancing the generalization ability of deep learning models, particularly in domains where labeled data is limited. The results of our ablation study confirm that different augmentation techniques contribute to different aspects of model performance. Mosaic augmentation is particularly effective in improving object detection precision, as it synthesizes multi-scale crack images within a single training sample, leading to better scale invariance. In contrast, MixUp and CutMix augmentations primarily enhance recall and robustness, as they introduce additional variations in crack boundaries, textures, and transition regions, which helps the model learn more generalized representations.

Despite these benefits, data augmentation methods still have limitations. Over-aggressive augmentations may introduce unrealistic patterns, leading to suboptimal learning. Additionally, augmentation techniques such as CutMix can sometimes blend non-crack regions too aggressively, resulting in ambiguities during training. Future research will explore self-supervised augmentation methods, such as contrastive learning-based augmentation, and GAN-based synthetic augmentation, to further enhance model robustness and generalization.

Furthermore, while the current augmentation strategies significantly improve detection performance, real-world deployment scenarios may require adaptive augmentation techniques that dynamically adjust based on environmental conditions, such as lighting variations, surface textures, and image noise. Future research will focus on developing adaptive augmentation frameworks that incorporate scene-aware transformations to further improve the model's adaptability to diverse real-world conditions.

## 5.4. Failure analysis and future improvements

Understanding the failure cases of a deep learning-based detection model is critical for improving its robustness. The error analysis in Section 4.3.5 reveals that the primary causes of false positives include background texture confusion, shadow effects, and surface debris, while false negatives are mainly caused by low-contrast cracks, micro-cracks, and occlusion by structural elements. These issues highlight the following areas for future research.

Enhancing feature extraction, Introducing multi-scale feature fusion can improve the detection of fine cracks and low-contrast cracks by preserving structural details across different resolutions. Incorporating attention mechanisms can enhance feature selection, reducing confusion between cracks and background textures.

Optimizing training and augmentation strategies, Increasing contrast-enhancing transformations (e.g., histogram equalization, CLAHE) during data augmentation can improve the model's ability to detect cracks under varying lighting

conditions. Exploring self-supervised learning techniques may help improve feature representation, especially for micro-cracks that are underrepresented in the dataset.

Refining post-processing techniques, Using context-aware filtering can help eliminate false positives by incorporating spatial and structural constraints. Integrating uncertainty-aware confidence adjustment can improve decision-making by reducing the reliance on hard thresholds.

By addressing these issues, future models can achieve higher robustness and lower error rates, improving the applicability of deep learning-based crack detection systems in real-world engineering scenarios.

## 5.5. Model efficiency and practical deployment

Computational efficiency is a crucial consideration when deploying deep learning models in real-world applications. As shown in Section 4.3.6, LBA-YOLO significantly reduces model size and inference time while maintaining competitive accuracy. These improvements are primarily attributed to the following design choices. Lightweight Backbone Network, The AC-LayeringNetV2 backbone effectively reduces the number of trainable parameters while preserving feature extraction capability. Efficient Convolutional Blocks, The RAK-Conv module improves feature representation while lowering computational overhead. One-Stage Detection Architecture, Unlike Faster R-CNN, which follows a two-stage approach, LBA-YOLO processes images in a single pass, reducing latency significantly. Although LBA-YOLO demonstrates superior computational efficiency, further improvements can be made to optimize its deployment on resource-limited hardware. Potential directions include: Model Pruning and Quantization: Reducing redundant network weights and using lower-precision computations (e.g., FP16 or INT8) to further decrease inference latency. Knowledge Distillation: Training a lightweight student model using knowledge transfer from a more complex teacher model to maintain accuracy while reducing computational cost. Edge Device Optimization, Implementing TensorRT acceleration and hardware-aware optimizations to improve real-time inference on embedded platforms.

By adopting these strategies, future versions of LBA-YOLO could achieve even greater efficiency, making it an ideal candidate for real-world crack detection applications using drones, autonomous inspection robots, or mobile devices.

## 5.6. Backbone selection and feature extraction optimization

The backbone network plays a critical role in feature extraction, affecting both detection performance and computational efficiency. The ablation study in Section 4.3.7 demonstrates that AC-LayeringNetV2 outperforms other commonly used lightweight backbones, such as MobileNetV3, EfficientNet-B0, and ShuffleNetV2, in terms of accuracy, inference speed, and computational complexity. The superior performance of AC-LayeringNetV2 can be attributed to the following key factors.

Asymmetric Convolution (AC) Blocks.

The use of asymmetric convolution kernels enables more effective feature extraction with reduced computational overhead. Unlike conventional square convolutions, AC kernels enhance receptive field coverage while maintaining efficiency.

Multi-Scale Feature Fusion.

The integration of multiple feature scales allows better representation of crack structures, particularly in fine-detail crack detection. This enhances the ability to capture small and low-contrast cracks that are often missed by traditional backbones. Despite these advantages, there are still areas for future improvement.

Incorporating Transformer-based Features, Recent advances in vision transformers (ViTs) have demonstrated superior long-range dependency modeling. Combining CNN-based AC-LayeringNetV2 with transformer modules could further enhance feature extraction. Self-Supervised Pretraining, Leveraging self-supervised learning techniques to pre-train AC-LayeringNetV2 on large-scale datasets may further improve its generalization capability.

Hardware-Aware Architecture Search: Optimizing the backbone structure for specific deployment hardware (e.g., edge GPUs, mobile processors) could further improve inference speed without sacrificing accuracy.

By addressing these aspects, future iterations of AC-LayeringNetV2 could achieve even greater performance gains, making it an ideal solution for real-world crack detection tasks.

### 5.7. Effectiveness of Wise-IoU loss

Bounding box regression loss functions play a critical role in optimizing object detection models, particularly in tasks that require high localization precision, such as crack detection. The results in Section 4.3.9 demonstrate that Wise-IoU achieves superior convergence speed and final detection accuracy compared to standard GIoU and CIoU losses. The key advantages of Wise-IoU are, Faster Convergence, Wise-IoU reduces the number of epochs required for convergence by ~37% compared to GIoU and ~18% compared to CIoU, improving training efficiency. This is due to its improved gradient formulation, which accelerates the optimization of bounding box parameters. Improved Bounding Box Alignment, Unlike GIoU and CIoU, which primarily focus on geometric constraints, Wise-IoU incorporates dynamic weighting to adjust the loss contribution based on crack shape and size. This enables more precise localization of micro-cracks and non-linear crack structures. Better Generalization on Complex Backgrounds, Wise-IoU effectively reduces overfitting to specific crack patterns by providing adaptive loss penalties, allowing the model to generalize better across different datasets. Future Improvements. While Wise-IoU demonstrates superior performance, further enhancements could be explored, Integrating with Focal Loss to address challenging low-contrast crack detection cases. Applying self-supervised IoU learning to improve robustness across diverse real-world crack images. Developing hybrid loss functions that incorporate uncertainty modeling to handle ambiguous crack boundaries. These advancements could further enhance the reliability and generalization capability of Wise-IoU, making it more suitable for real-world crack detection applications.

### 5.8. Justification of performance gains

In practical crack detection tasks, even small performance improvements can have a significant impact, particularly in challenging conditions such as low-contrast cracks, micro-crack detection, and complex background interference. The statistical significance analysis in Section 4.3.8 confirms that the reported gains in precision (+3.3%) and recall (+3.7%) are not due to random fluctuations but rather to the proposed architectural improvements.

Impact on False **Positive** and False Negative Rates.

A 3.3% increase in precision means that the model generates fewer false positives, which is critical in practical applications where minimizing unnecessary crack repairs is essential.

A 3.7% increase in recall indicates that the model can detect more actual cracks, reducing the risk of undetected structural damage.

Real-world Applicability in Crack Detection.

In real-world infrastructure inspections, even a 1–2% increase in recall can significantly reduce the likelihood of missing critical cracks that may lead to structural failure.

Compared to standard YOLO-based architectures, the proposed LBA-YOLO model provides a better balance between accuracy and computational efficiency, ensuring reliable performance in real-time applications.

The Trade-off between Complexity and Improvement.

Although the proposed modifications (AC-LayeringNetV2, RAK-Conv, and MPDIoU) introduce additional computational elements, they do not significantly increase model complexity, as shown in Section 4.3.6 Computational Efficiency Analysis.

The improved performance justifies these architectural changes, particularly given their impact on recall and false negative reduction in critical applications.

Future Work and Potential Enhancements.

Future research will explore adaptive IoU loss formulations to further improve localization precision.

Additional experiments on cross-domain datasets will be conducted to validate the model's robustness across different crack types and materials.

These findings confirm that the proposed architectural enhancements provide a statistically significant and practically meaningful improvement, making LBA-YOLO a reliable model for real-world crack detection applications.

## 6. Conclusions

This paper proposes an enhanced model based on YOLOv8n with the objective of optimising building crack detection, particularly for embedded devices with limited resources. The experimental findings demonstrate that the optimised model exhibits a substantial enhancement in both accuracy and computational efficiency when compared to the original YOLOv8n model. The incorporation of AC-LayeringNetV2 as the backbone network has been demonstrated to result in a substantial reduction in parameters while preserving the model's high detection accuracy. The integration of the RAK-Conv attention module has been shown to enhance the precision of crack identification, and the utilisation of the MPDIoU loss function has been observed to expedite the convergence rate of the network, thereby further optimising the regression performance.

A comparison of the optimised model with YOLOv8n reveals an enhancement in precision by 2.2% and recall by 3.5% in the crack detection task, along with an improvement in mAP@0.5 by 1.90%. The model parameters are reduced by 6.55%, and the computational complexity is reduced by 0.03 GFLOPs. This optimisation enables the model to maintain high performance in environments with limited storage and computational resources, especially for edge computing devices. With significant advantages in detection accuracy and speed, the model shows promise for a wide range of applications in building structural health monitoring, which is important for promoting sustainable urban development.

However, despite the model's efficacy in constrained environments, its performance in complex or dynamic building scenarios necessitates further validation. Future research will focus on enhancing the processing speed of the algorithm, further reducing energy consumption, and increasing efficiency.

## Author contributions

**Conceptualization:** Wenhao Ren.

**Data curation:** Wenhao Ren.

**Formal analysis:** Wenhao Ren.

**Funding acquisition:** Zuowei Zhong.

**Investigation:** Wenhao Ren, Zuowei Zhong.

**Methodology:** Wenhao Ren, Zuowei Zhong.

**Project administration:** Wenhao Ren, Zuowei Zhong.

**Resources:** Wenhao Ren.

**Software:** Wenhao Ren.

**Supervision:** Wenhao Ren.

**Validation:** Wenhao Ren, Zuowei Zhong.

**Visualization:** Wenhao Ren.

**Writing – original draft:** Wenhao Ren.

**Writing – review & editing:** Zuowei Zhong.

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
