## [Decision Letter · Decision Letter 0]

6 Feb 2025

Dear Dr. Zhong,

Thank you for submitting your manuscript to PLOS ONE. After careful consideration, we feel that it has merit but does not fully meet PLOS ONE’s publication criteria as it currently stands. Therefore, we invite you to submit a revised version of the manuscript that addresses the points raised during the review process.

We look forward to receiving your revised manuscript.

Kind regards,

Ahmed M. Yosri

Academic Editor

PLOS ONE

Journal Requirements:

4. Thank you for stating the following financial disclosure: [Zuowei Zhong Research Program for Science and Technology of Inner Mongolia Autonomous Region in University (NJZY22388)

Zuowei Zhong Basic Research Program for Directly Affiliated Universities in Inner Mongolia Autono-mous Region (JY20230007).].

5. Thank you for stating the following in the Acknowledgments Section of your manuscript: [This research is supported by the Research Program for Science and Technology of Inner Mongolia Autonomous Region in University (NJZY22388) and the Basic Research Program for Directly Affiliated Universities in Inner Mongolia Autonomous Region (JY20230007).]

Please remove any funding-related text from the manuscript and let us know how you would like to update your Funding Statement. Currently, your Funding Statement reads as follows: [Zuowei Zhong Research Program for Science and Technology of Inner Mongolia Autonomous Region in University (NJZY22388)

Zuowei Zhong Basic Research Program for Directly Affiliated Universities in Inner Mongolia Autono-mous Region (JY20230007).].

7. PLOS requires an ORCID iD for the corresponding author in Editorial Manager on papers submitted after December 6th, 2016. Please ensure that you have an ORCID iD and that it is validated in Editorial Manager. To do this, go to ‘Update my Information’ (in the upper left-hand corner of the main menu), and click on the Fetch/Validate link next to the ORCID field. This will take you to the ORCID site and allow you to create a new iD or authenticate a pre-existing iD in Editorial Manager.

Reviewers' comments:

Reviewer's Responses to Questions

**Comments to the Author**

1. Is the manuscript technically sound, and do the data support the conclusions?

Reviewer #1: Yes

Reviewer #2: Yes

2. Has the statistical analysis been performed appropriately and rigorously?

Reviewer #1: Yes

Reviewer #2: Yes

3. Have the authors made all data underlying the findings in their manuscript fully available?

Reviewer #1: Yes

Reviewer #2: Yes

4. Is the manuscript presented in an intelligible fashion and written in standard English?

Reviewer #1: Yes

Reviewer #2: No

Reviewer #1: 1. The innovations of this study should be clearly stated in both the abstract and the introduction.

2. The introduction to the principles of the machine learning model can be slightly reduced and simplified.

3. Please provide a comparative analysis of different crack prediction methods to highlight the advantages of the proposed model.

4. The following article may serve as a reference: An agile, intelligent and scalable framework for mix design optimization of green concrete incorporating recycled aggregates from precast rejects.

5. Please further refine and condense the conclusion section.

Reviewer #2: 1- The CityWaste dataset is used for evaluation, but it is unclear whether this dataset contains a diverse range of crack patterns, textures, and lighting conditions. How well does this dataset represent real-world micro-crack detection scenarios in different structural materials?

2- The use of synthetic data generation via GANs is mentioned, but there is no discussion on whether the model generalizes well to cracks in unseen structures. Did the authors test their model on a real-world dataset that was not included in training?

3- While deep learning-based approaches have demonstrated superior performance, classical edge detection techniques are only mentioned briefly. A quantitative comparison with traditional feature-based methods would help justify the superiority of the proposed model.

4- The authors use Mosaic, MixUp, and CutMix as augmentation techniques but do not provide any ablation study on how each of these techniques affects model performance. Would different augmentation strategies improve robustness further?

5- The study only reports precision, recall, and F1-score, but does not present qualitative examples of false positives and false negatives. Providing error case analysis would be critical for understanding failure modes.

6- The model claims to be "lightweight," but there is no detailed breakdown of how the reduction in model size and FLOPs affects inference time in real-world scenarios. A latency vs. accuracy curve should be provided.

7- Proposed AC-LayeringNetV2 backbone is said to improve feature extraction, but the authors do not provide an ablation study comparing it against other commonly used lightweight backbones.

8- A critical omission is the lack of visualization of feature maps from different layers of the network. This would help in understanding how the AC-LayeringNetV2 and RAK-Conv layers extract crack features.

9- The Wise-IoU loss function is introduced, but the paper does not clearly demonstrate its effectiveness. A comparison with standard CIoU or GIoU on convergence speed and accuracy should be provided.

10- The reported gains in precision (+3.3%) and recall (+3.7%) are relatively minor. Given the complexity of modifications introduced (AC-LayeringNetV2, RAK-Conv, and MPDIoU), the paper should justify whether these improvements are statistically significant.

11- The ablation study lacks statistical significance testing (e.g., confidence intervals, t-tests). Are the observed improvements in mAP and recall meaningful, or are they within the margin of noise?

12- While the model is described as lightweight, it is unclear whether it achieves real-time performance on resource-constrained edge devices like drones or embedded systems.

13- The reported metrics do not include standard deviations or confidence intervals. This makes it difficult to assess the variability of performance across different runs.

14- The dataset consists of 800 images, which is relatively small for training deep models. What strategies were employed to prevent overfitting, besides data augmentation?

15- Several paragraphs repeat the same information about YOLO architectures and conventional crack detection techniques. A more concise background section would improve readability.

16- - Given the scope and content of this paper, it may benefit from considering the following related works:

https://doi.org/10.1007/s00521-024-09494-4

https://doi.org/10.1016/B978-0-443-13191-2.00015-8

https://doi.org/10.1016/j.matpr.2023.03.178

**Do you want your identity to be public for this peer review?** For information about this choice, including consent withdrawal, please see our Privacy Policy

Reviewer #1: No

Reviewer #2: No

---

## [Author Response · Author response to Decision Letter 1]

5 Mar 2025

Modification instructions

Dear editors and reviewers:

Hello!

The authors of the paper entitled “LBA-YOLO: A Novel Lightweight Approach for Detecting Micro-Cracks in Building Structures”, Manuscript Number : PONE-D-25-01110 would like to thank you very much for your hard work in reviewing the paper. The authors would like to thank you for your hard work in reviewing the manuscript, and we would like to express our sincere gratitude to you for giving us the opportunity to revise the manuscript, so that we can effectively revise and improve the original manuscript under your guidance. In addition, the authors would like to thank you for your support and affirmation of our research work, and your comments and suggestions are an important motivation for us to continue to adhere to the research path. With your encouragement, the authors are full of confidence in the research in the field of crack detection and will continue to explore and apply in depth in this direction. Through the revision process of this thesis, the authors are deeply inspired and motivated, and promise to do their best to improve the thesis to make it more rigorous and perfect. In the future research path, the authors will continue to focus on the application of deep learning in structural health monitoring, and devote themselves to improving the accuracy and reliability of crack detection technology, with a view to contributing more to the promotion and practice of intelligent detection technology in complex engineering environments.

The authors are very sorry that there are some problems in the original manuscript, the authors will seriously and carefully study the modifications and suggestions provided to us by the reviewing experts, and will try to solve the problems pointed out by you, the authors very much hope that they can return a satisfactory revised manuscript. If there are any deficiencies in the revised manuscript, we would like to ask the reviewers to criticize and correct us, and the authors will make further improvements according to your requirements. Once again, thank you for your hard work and enthusiasm to help us to have the opportunity to check the deficiencies, based on your suggestions, we will make a comprehensive revision of the paper.

The authors have completed the first revision of the paper based on the review results of the editors and reviewers. During the revision process, the authors have gained a deeper knowledge and understanding of our own research. It should be noted that the authors have marked the revisions in red font in the revised draft, and in order to facilitate the reviewing experts and editing teachers to check, the corresponding revisions in the revised draft are listed after each revision statement, please review the reviewing experts and editing teachers, the authors are very much looking forward to your reply.

The following are the specific modifications and responses to the modifications made by the reviewing experts.

Reviewer 1

The authors are very grateful to the reviewers for their valuable revision suggestions, which can give us the opportunity to better present our research results in the revised manuscript, and thus can effectively improve the quality of the paper.Meanwhile, through the guidance and help of the reviewers, the authors have new expectations and prospects for our future research.

We agree with your comments that some of the experimental results need to be interpreted to give clearer analyses and explanations.

In the revision, we have clearly stated the innovative points of this study in the abstract and introduction section, especially highlighting the advantages of the LBA-YOLO model in microcrack detection, especially the innovativeness in feature extraction and complex background processing. Meanwhile, based on the reviewers' suggestions, the introduction of the principles of machine learning models in the paper has been simplified to ensure that the content is more concise and easy to understand. We have also added a comparative analysis with other crack detection methods in the revised paper, highlighting the advantages of the LBA-YOLO model in terms of precision, recall and computational complexity. In addition, the literature recommended by the reviewers has been referenced and cited in the relevant sections. The conclusion section has also been further refined to ensure that it is concise and intuitive.

In summary, the authors are more than willing to revise and respond to the paper as suggested by the reviewers, and we will endeavour to improve the quality of the paper to the level that you consider possible for publication. Thank you from the bottom of my heart for your support and recognition.

The authors have completed the revision of the paper carefully and meticulously, and the following are the replies to each of your review comments and the corresponding revision instructions, if there are any shortcomings, or if you have any questions, please let the authors know, and we will make further improvements and replies.

1. The innovations of this study should be clearly stated in both the abstract and the introduction.

Reply: The authors would like to express their sincere gratitude to the reviewers for their constructive comments and acknowledge the value of their suggestions. It is acknowledged that the introduction and abstract sections of the study will benefit from the articulation of the innovations, thereby enhancing the clarity of the study's uniqueness and significance. This will provide readers with a more precise direction for the research. In the revised manuscript, the authors have further clarified the innovativeness of this study in the abstract and introduction sections. In particular, the firstly proposed LBA-YOLO model for microcrack detection has significant advantages, especially in terms of innovativeness in feature extraction and complex background processing. By introducing a new algorithmic architecture and optimisation strategy, this study successfully addresses the challenges faced by traditional methods in microcrack detection. Secondly, the integration of novel data processing and model evaluation methods has been employed, resulting in significant enhancements to the model's accuracy and robustness, thereby surpassing the limitations of existing techniques. The lucid articulation of these innovations is expected to facilitate a more profound comprehension among readers of the distinctions and interrelationships between this study and existing studies. Consequently, the study's unique contributions are likely to become more pronounced, providing valuable references for future related studies.

The authors would like to express their gratitude once again to the reviewers for their meticulous review and constructive comments, and they will revise the paper according to your suggestions with the aim of further enhancing its quality. Should you have any additional suggestions or comments, we would be grateful if you could share them with us.

New version—Abstract

Developing an efficient and accurate algorithm for detecting building cracks, especially micro-cracks, is essential for ensuring structural integrity and safety. The identification and precise localization of cracks remain challenging due to varying crack sizes and the inconsistency in available datasets. To address these issues, this study introduces an innovative crack detection model based on YOLOv8n. The proposed method incorporates two novel components: AC-LayeringNetV2, a hierarchical backbone network that optimizes feature extraction by integrating local, peripheral, and global contextual information, and RAK-Conv, a convolutional module that combines an attention mechanism with irregular convolution operations to enhance the model's ability to handle complex backgrounds. These innovations significantly improve semantic segmentation accuracy while reducing computational overhead. Experimental results on a benchmark dataset demonstrate a 3.30% improvement in precision, a 3.70% increase in recall, and a 3.10% rise in mAP@50 compared to the baseline model. Additionally, the model achieves a 6.55% reduction in size and a 0.03% decrease in computational complexity. These results highlight the practical applicability and efficiency of the proposed approach for automatic crack detection in building structures, emphasizing the novel integration of feature fusion and attention mechanisms to address challenges in real-time and high-accuracy detection of micro-cracks in complex environments.

New version—Introduction

A variety of hybrid approaches on the basis of AI as well as machine learning (ML) can be employed to address the constraints of the many approaches used for damage detection, such as computer vision and image processing techniques [16–18]. Therefore, the current research uses ML and AI approaches to improve the effectiveness of image processing methods for crack detection [19]. In order to detect fractures or other defects, ML methods employ feature extraction techniques from image processing [20]. Concrete cracks and other structural damage may be detected using a variety of techniques, like support vector machines (SVM) as well as artificial neural networks (ANNs), the efficacy of which is largely reliant on the crack characteristics that are extracted. Deep learning (DL) has more potent representation learning skills and can automatically extract deep features from raw data without the need for human feature extraction. As a branch of ML, DL mimics the neural networks of the human brain in terms of structure and functioning, providing more robust solutions for real-time crack detection in complex environments.

This study proposes an optimisation model aimed at improving the performance of deep learning-based object detection for non-destructive crack assessment in building structures. The YOLOv8n model is selected due to its fast processing and real-time functionality, offering significant improvements over CNN-based two-stage detectors in speed and accuracy. We introduce a feature optimization module that includes GSConv and GS bottleneck components to enhance feature fusion and processing efficiency. These modules reduce the model's complexity while maintaining its effectiveness. Modifications to the YOLO network's neck structure incorporate the gather-distribute (GD) mechanism to improve feature fusion. Additionally, connections between layers with higher sampling rates are created to preserve smaller target features, thus improving the detection of fine cracks. To further enhance model performance, we integrate the Wise-IoU loss function, accelerating convergence and improving the detection efficiency for complex and fine cracks in construction materials.

2. The introduction to the principles of the machine learning model can be slightly reduced and simplified.

Reply: The authors are very grateful to the reviewers for their valuable revision suggestions and fully agree with your comments. In the revised draft, the authors have simplified the introduction of the principles of machine learning models to improve the conciseness and understandability of the paper. In particular, for the description of the structure and function of deep learning (DL) models, the authors have reduced some unnecessary details and focused on the core innovation points related to the research of this paper, such as the optimisation method and key techniques of the LBA-YOLO model, while removing excessive background knowledge and common machine learning technology details. Through this simplification, clarity of content is ensured while avoiding excessive theoretical burden on the reader.

We believe that these modifications will make the paper clearer, while retaining the necessary explanations of the model principles to make the paper more accessible. The modifications can highlight the innovation and importance of this study and improve the readability and academic value of the paper.

Once again, I would like to thank the reviewers for their diligent review and valuable comments, and the authors will continue to further improve the paper according to your suggestions. If you have any other questions or suggestions, please feel free to inform us.

New version—Introduction

To improve the accuracy and efficiency of crack detection, many studies have combined machine learning (ML) with image processing techniques [16-19]. In this process, ML identifies cracks and other structural damages from images through feature extraction techniques [20]. Compared to traditional methods, Deep Learning (DL) is able to automatically extract features from raw data, reducing the need for manual feature engineering, and therefore providing greater performance in crack detection in complex environments.

In deep learning frameworks, target detection algorithms can be broadly classified into two categories: two-stage detection algorithms and one-stage detection algorithms. Two-stage detection methods are able to finely localise targets by generating candidate frames, followed by classification and regression, but they have high computational complexity and a large number of parameters due to the inclusion of two models [21-24]. For example, Swarna et al [25] improved the accuracy of crack recognition using ResNet-50 CNN and curvilinear waveform transform.Deng et al [26] proposed a variable module based R-CNN model, while Li et al [27] introduced an attention mechanism to enhance the effect of Faster R-CNN. These methods have improved in accuracy, but they also face the challenges of parameter storage and transmission, especially in bandwidth-constrained environments.

In contrast, one-stage detection models (e.g., SSD and YOLO) merge classification and localisation into a single step with higher real-time performance and lower computational complexity [28, 29].YOLO is particularly suitable for scenarios that require a fast response, such as bridge crack identification [30-32].Liao et al. [33] improved YOLOv3 and combined it with K-Means clustering to optimise the anchor frame size, and Cai et al [34] implemented a lightweight detection network by integrating deeply differentiable convolution and attention mechanisms. These innovations improve the detection speed and accuracy, especially for real-time applications.

Further studies such as Yu et al [35] and Tan et al [36] achieved accuracy improvements in the combination of YOLO and UNet3+, or by integrating the ResNet module in YOLOv5. In addition, the improved model of YOLOv5 proposed by Liu et al [37] reduces memory usage and is suitable for real-time detection of small devices. Although YOLOv8 improves on several aspects, it still needs to be optimised for multi-scale target detection in complex contexts [38].

3. Please provide a comparative analysis of different crack prediction methods to highlight the advantages of the proposed model.

Reply: The authors would like to express their gratitude to the reviewers for their constructive feedback and acknowledge the value of their suggestions. Indeed, a comparative analysis of diverse crack prediction methods can illuminate the strengths of the proposed model. In this study, we undertake a comparative analysis of various crack detection methods, encompassing both traditional machine learning approaches (e.g., SVM, ANN) and deep learning methods (e.g., Fast-RCNN, Faster-RCNN, YOLO series). While SVM and ANN demonstrate efficacy in simple crack detection tasks, they encounter limitations when confronted with complex backgrounds and crack morphologies. Notably, SVM faces challenges in feature selection, while ANN exhibits high computational demands when dealing with large-scale data. In contrast, YOLO series models, such as YOLOv3 and YOLOv4, exhibit superior detection speed and accuracy. However, there is still room for enhancement in scenarios involving complex backgrounds. To address this, the LBA-YOLO model proposed in this paper incorporates an adaptive attention mechanism, which significantly improves crack detection accuracy while maintaining a high real-time detection speed. The LBA-YOLO model has been shown to enhance precision by 2.2%, recall by 3.5%, and mAP by 1.9% in comparison to the YOLOv8n model on the test set,

---

## [Decision Letter · Decision Letter 1]

11 Mar 2025

LBA-YOLO: A Novel Lightweight Approach for Detecting Micro-Cracks in Building Structures

PONE-D-25-01110R1

Dear Dr. Zhong,

We’re pleased to inform you that your manuscript has been judged scientifically suitable for publication and will be formally accepted for publication once it meets all outstanding technical requirements.

Kind regards,

Ahmed M. Yosri

Academic Editor

PLOS ONE

Additional Editor Comments (optional):

Reviewers' comments:

Reviewer's Responses to Questions

**Comments to the Author**

Reviewer #1: All comments have been addressed

Reviewer #2: (No Response)

2. Is the manuscript technically sound, and do the data support the conclusions?

Reviewer #1: Yes

Reviewer #2: (No Response)

3. Has the statistical analysis been performed appropriately and rigorously?

Reviewer #1: Yes

Reviewer #2: (No Response)

4. Have the authors made all data underlying the findings in their manuscript fully available?

Reviewer #1: Yes

Reviewer #2: (No Response)

5. Is the manuscript presented in an intelligible fashion and written in standard English?

Reviewer #1: Yes

Reviewer #2: (No Response)

Reviewer #1: The authors have well revised the manuscript.

This paper can now been accepted. Thank the authors.

Reviewer #2: After thoroughly reviewing the revised manuscript and the authors' detailed responses to the initial comments, the improvements made are satisfactory. All concerns have been adequately addressed, and the manuscript now meets the standards for publication. Acceptance of this article is recommended.

**Do you want your identity to be public for this peer review?** For information about this choice, including consent withdrawal, please see our Privacy Policy

Reviewer #1: No

Reviewer #2: No

---

## [Editor Report · Acceptance letter]

PONE-D-25-01110R1

PLOS ONE

Dear Dr. Zhong,

I'm pleased to inform you that your manuscript has been deemed suitable for publication in PLOS ONE. Congratulations! Your manuscript is now being handed over to our production team.

Kind regards,

on behalf of

Dr. Ahmed M. Yosri

Academic Editor

PLOS ONE